# Revealing disorder parameter and deformation electron density using electron diffraction

Weixiao Lin[1,2,6], Zefan Xue[1,2,6], Wenjun Cui[1,2], Andreas Kulovits[3], Hao Ren[4], Wen Zhao[4], Jinsong Wu[1,2], Gustaaf Van Tendeloo[2,5], Jörg Wiezorek[3] & Xiahan Sang[1,2] ✉

Local disorders of lattice, charge, orbital, and spin perturbate the electron density distribution in materials, profoundly influencing their properties. Consequently, experimental determination of local electron density offers a powerful, universal approach to probe such disorder. Although quantitative convergent beam electron diffraction (QCBED) is widely employed for electron density measurements in ordered crystals, its applicability to disordered structures, where the translational symmetry of the electrostatic potential is broken, remains uncertain. Here, a multi-beam off-zone axis CBED technique combined with a coherent potential approximation in Bloch wave formalism is used to simultaneously determine chemical disorder parameters, deformation electron density $\Delta\rho^{EXP}$, and Debye-Waller factors (DWF) in both chemically-ordered $L1_0$ FePd and chemically-disordered $\gamma$-phase FePd solid solution. The CBED results reveal that chemical disordering significantly increases DWFs while having a negligible impact on $\Delta\rho^{EXP}$. Density functional theory calculations on supercells with randomly distributed Fe and Pd atoms support these experimental findings. This work validates QCBED as a robust method for quantifying local disorder parameters in chemically disordered systems, bridging a critical gap in the characterisation of disordered materials.

Structural disorders, including chemical composition, charge, orbital, and spin, break the translational symmetry of periodic crystals[1]. These disorders are pervasive in diverse materials, including two-dimensional materials[2], oxides[3,4], and alloys[5,6], where they profoundly influence critical properties such as ferroelectricity[3], thermoelectric efficiency[7,8], proton conductivity[9,10], and mechanical behaviour[6,11]. Characterising disorders at the (sub)nanoscale has become increasingly important for nanomaterials[12], heterointerfaces[13], and surfaces[14]. While atomic-resolution imaging and spectroscopy techniques offer insights into local chemical disorder, probing local disorders related to

charge[15], orbital, or spin[16] remains challenging when relying solely on sampling ordered-sensitive Bragg diffraction intensities. Crucially, all forms of disorder perturb electronic structures and atomic displacements, collectively modifying bonding and electron density distribution. Consequently, precise determination of local electron density emerges as a universal pathway to unravelling local disorders.

In crystalline materials, electron density can be experimentally determined using X-ray diffraction (XRD) or quantitative convergent beam electron diffraction (QCBED) refinements[17–20]. While XRD provides ensemble-averaged structural information, QCBED utilises a

[1]State Key Laboratory of Advanced Technology for Materials Synthesis and Processing, Wuhan University of Technology, Wuhan, China. [2]Nanostructure Research Center, Wuhan University of Technology, Wuhan, China. [3]Department of Mechanical Engineering and Materials Science, University of Pittsburgh, Pittsburgh, PA, USA. [4]School of Materials Science and Engineering, China University of Petroleum (East China), Qingdao, China. [5]EMAT (Electron Microscopy for Materials Science), University of Antwerp, Antwerp, Belgium. [6]These authors contributed equally: Weixiao Lin, Zefan Xue. ✉e-mail: xhsang@whut.edu.cn

nanometre-sized focused electron probe, often integrated with transmission electron microscope (TEM) imaging, to achieve nanoscale or even atomic-scale spatial resolution. This capability allows for direct investigation of local disorders in specific regions. Recent decades have seen significant advancements in QCBED methodologies, including systematic-row[17,18,21], zone-axis[22–24], and multi-beam off-zone axis (MBOZA) approaches[19,20,25–28]. These techniques enable precise measurement of bonding-affected low-order X-ray structure factors ($F_g^X$), which encode subtle electron redistribution arising from chemical interactions. The experimental deformation electron density map ($\Delta\rho^{EXP}(r)$), reflecting the influence of bonding, is the Fourier transform of the difference between experimentally measured ($F_g^{EXP}$) and independent atom model (IAM) structure factors ($F_g^{IAM}$) that ignores interatomic bonding effects:

$$\Delta\rho^{EXP}(r) = \sum_g \left(F_g^{EXP} - F_g^{IAM}\right)e^{-2\pi i g \cdot r} \qquad (1)$$

While QCBED has been used to determine $\Delta\rho^{EXP}(r)$ in ordered materials such as oxides[17], elemental metals[19], and intermetallics[29], with good agreement between experimental results and density functional theory (DFT) predictions[30], investigations into $\Delta\rho^{EXP}(r)$ of disordered materials remain limited[31] due to three key challenges. First, the disorder parameter may vary spatially. Simultaneous determination of the disorder parameter and $\Delta\rho^{EXP}(r)$ from CBED patterns remains challenging. Second, the Debye-Waller factors (DWF), reflecting local atomic displacements and affecting $\Delta\rho^{EXP}(r)$, must be determined concurrently. Third, both Bloch wave formalism and DFT simulations inherently presume long-range periodicity in crystal potentials[32]. Disordered structures necessitate approximating an effective periodic potential, an assumption that remains unvalidated and may introduce systematic errors in interpreting $\Delta\rho^{EXP}(r)$.

Here, $\Delta\rho^{EXP}(r)$ of chemically disordered face-centred cubic (FCC) FePd solid solution and L1$_0$ ordered FePd are measured using the MBOZA QCBED method. The MBOZA condition enhances beam-sample dynamic interactions, enabling simultaneous determination of DWF and $F_g^X$ from nanoscale regions[20,28,29,33–35]. A coherent potential approximation (CPA), assuming an averaged periodic potential[36], is employed in Bloch wave calculations for disordered structures. The QCBED results highlight its sensitivity to the disorder parameter,

revealing larger DWFs in FCC FePd than in L1$_0$ ordered FePd due to local lattice distortions[19,20,33], consistent with DFT calculations using supercells containing randomly distributed Fe and Pd atoms. While both phases exhibit similar averaged low-order $F_g^X$ values, local bonding and charge transfer are found to vary with nearest-neighbour configurations. This work underscores the potential of QCBED and DFT for probing disorder parameters, $\Delta\rho^{EXP}(r)$, and bonding in disordered materials.

## Results and discussion
### MBOZA QCBED measurement of the disorder parameter $\eta$

The L1$_0$ ordered FePd phase possesses a tP2 tetragonal unit cell with Fe and Pd atoms occupying (0, 0, 0) and (0.5, 0.5, 0.5) positions, respectively (Fig. 1a). Chemical disorder arises from the mutual exchange of Fe and Pd atoms on each other's sites. This disorder for FePd is quantified by the parameter $\eta = 2 \times Occ.$ (Fe$_{Pd}$), where $Occ.$ (Fe$_{Pd}$) represents the partial site occupancy of Fe atoms at the Pd site. The disorder parameter $\eta$ is inversely related to the conventional long-range order parameter $S$ for Fe at its native site (0, 0, 0) by $\eta = 1 - S$ (Supplementary Note 1). Here, $\eta = 0$ indicates perfect chemical order, while $\eta = 1$ represents complete chemical disorder. Under MBOZA conditions, the crystal is tilted slightly from a zone axis to satisfy Bragg conditions for multiple diffracted beams simultaneously (Fig. 1b). This geometry ensures precise intersection of the Ewald sphere with reciprocal lattice points, enabling simultaneous excitation of multiple diffracted beams, and producing both bound and anti-bound Bloch waves that interact with core electrons and bonding electrons, respectively[35]. While MBOZA CBED has demonstrated sensitivity to changes in low-order $F_g^X$ and DWFs[33,35], its sensitivity to variations in $\eta$ has not been systematically investigated.

For tP2 FePd, the X-ray structure factors $F_g^X$ are described by the following equations (Supplementary Note 2):

$$F_g^X = f_{Fe}(s)e^{-B_{Fe}s^2} + f_{Pd}(s)e^{-B_{Pd}s^2}, h+k+l \text{ is even} \qquad (2)$$

$$F_g^X = (1-\eta)\left(f_{Fe}(s)e^{-B_{Fe}s^2} - f_{Pd}(s)e^{-B_{Pd}s^2}\right), h+k+l \text{ is odd} \qquad (3)$$

where $s$ is $\sin\theta/\lambda$ or $|g|/2$, and $B_{Fe}$ and $B_{Pd}$ are DWFs for Fe and Pd, respectively. $f_{Fe}$ and $f_{Pd}$ are $s$-dependent IAM scattering factors that

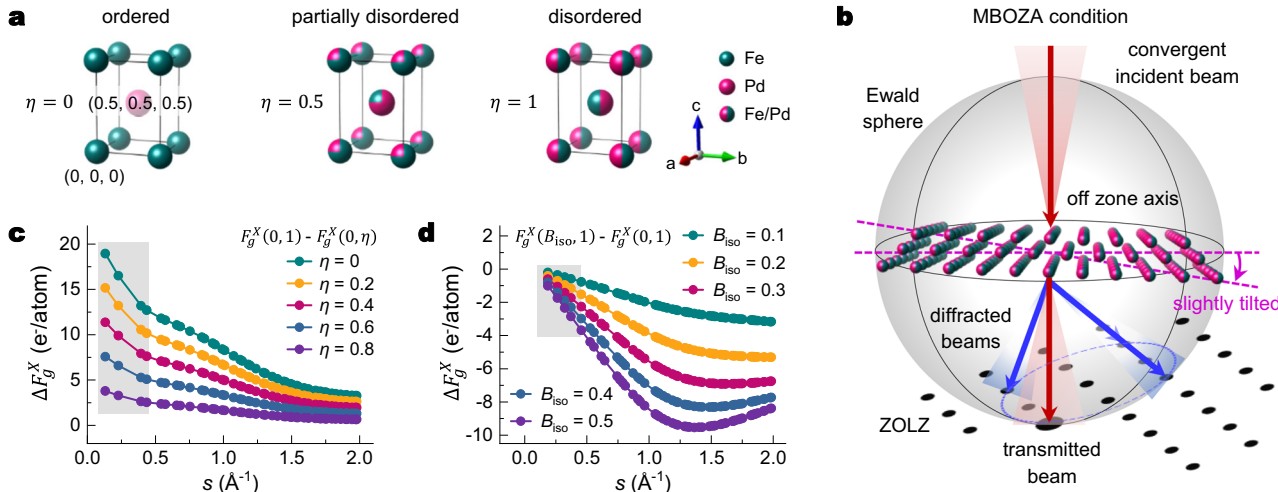

**Fig. 1 | Crystal structure and X-ray structure factors of tP2 FePd. a** Crystal structure of the tP2 FePd unit cells with disorder parameters 0, 0.5, and 1. **b** Schematic of a multi-beam off-zone axis (MBOZA) condition showing beam-sample orientation and the intersection (blue dashed line) between zeroth-order Laue zone (ZOLZ) and the Ewald sphere. **c** $F_g^X(0, 1) - F_g^X(0, \eta)$ ($h+k+l$ odd, $B_{iso}$

fixed at 0) as a function of $s$ for $\eta = 0, 0.2, 0.4, 0.6,$ and 0.8. **d** $F_g^X(B_{iso}, 1) - F_g^X(0, 1)$ ($h+k+l$ even, $\eta$ fixed at 1) as a function of $s$ for $B_{iso} = 0.1, 0.2, 0.3, 0.4,$ and 0.5. Grey rectangular shadows in (**c**, **d**) highlight the behaviour of low-order $F_g^X$ from (001) to (222) with $s$ ranging from 0.131 to 0.455 Å$^{-1}$. Source data are provided as a Source Data file.

have been calculated for atoms of the periodic table[37,38]. For simplicity, an averaged isotropic DWF ($B_{iso}$) is assumed with $B_{Fe} = B_{Pd} = B_{iso}$. To visualise the influence of $\eta$ on $F_g^X(B_{iso}, \eta)$, the difference between $F_g^X(0, 1)$ and $F_g^X(0, \eta)$ is plotted with $B_{iso}$ fixed at 0 (Fig. 1c). $F_g^X(0, 1)$ represents the structure factor of the fully disordered ($\eta = 1$) and static ($B_{iso} = 0$) tP2 cell. As $\eta$ decreases, $\Delta F_g^X$ with $h + k + l$ odd increases for all $s$, with a maximal deviation of 18.942 at $s = 0.131$ Å$^{-1}$ (Fig. 1c). Similarly, the influence of $B_{iso}$ is visualised by plotting the difference between $F_g^X(B_{iso}, 1)$ and $F_g^X(0, 1)$ with $\eta$ fixed at 1 (Fig. 1d). Here, $\Delta F_g^X$ shows greater sensitivity to the change in $B_{iso}$ at large $s$, with a maximal deviation of 9.531 at $s = 1.365$ Å$^{-1}$ (Fig. 1d). Thus, $\eta$ and DWF predominantly impact low-order and high-order structure factors, respectively. Moreover, the variation in $\eta$ produces a more pronounced change in $F_g^X$, indicating that MBOZA conditions offer sufficient sensitivity for measuring the disorder parameter.

For tP2 FePd, an MBOZA condition close to the [100] zone axis with (002), (020), and (022) diffraction spots on the Ewald sphere was selected (Fig. 2a). The blue dashed line in Fig. 2a delineates the intersection of the Ewald sphere with the zeroth-order Laue zone (ZOLZ). Energy-filtered CBED patterns were then acquired at −170 °C from an L1$_0$ FePd sample (Fig. 2b) and a supposedly disordered FCC FePd sample (Fig. 2c, d), cooled using a cold stage. The (0$kl$)-type discs with $k + l$ odd are observable in the ordered pattern (Fig. 2b, black arrow). Selected area electron diffraction (SAED) data from the disordered sample indicates that the intensity of these ordered reflections ($h + k + l$ odd) is only 0.1% of the main reflections (Supplementary Fig. 1). Given that the ordered discs in the two disordered CBED patterns display intensities comparable to the background (Supplementary Fig. 2), only the main reflection discs, (000), (01$\bar{1}$), (020), (002), (022) and (013), are included for QCBED refinement.

QCBED refinement was performed on the three experimental CBED patterns using the MBFIT software[22,23], minimising the difference between observed pixel intensities ($I_i^{obs}$) and calculated pixel intensities ($I_i^{cal}$) within the selected diffraction discs (Fig. 2e–h and Supplementary Figs. 3–5). Each pattern comprises a substantial dataset (71,739, 84,017, and 44,519 pixels, respectively), ensuring robust statistical refinement. Preprocessing of $I_i^{obs}$ involves background subtraction as determined from inter-disc regions, and linear distortion correction based on disc positions (Supplementary Fig. 6). For $I_i^{cal}$, overall blurring effects, including the point spread function (PSF) and local lattice variation, are modelled by convolution with a two-dimensional (2D) Gaussian distribution or Voigt distribution (Supplementary Figs. 7–9 and Supplementary Note 3). Beam damage was deemed negligible, as the maximum energy transferred from the 200 kV electron beam to Fe (9.4 eV) and Pd (4.9 eV) atoms remains below their displacement thresholds (-22.8 eV, Supplementary Note 4).

The theoretical CBED intensity was calculated using Bloch wave formalism within the refinement programme[39]. A CPA approximation was used, assuming that each lattice site has the same averaged potential related to the disorder parameter $\eta$. For simplicity, an averaged isotropic DWF, $B_{iso}$, was assigned for both Fe and Pd atoms. The initial values of $F_g^X$ were set to IAM values[37], providing a good approximation for high-order $F_g^X$ that are negligibly influenced by bonding. Absorption factors were calculated using the method described in ref. 40. For each CBED pattern, fifty refinements were performed with $\eta$ fixed at values ranging from 0 to 1 with a step size 0.02. Simultaneously, refinement involved relaxing fitting parameters such as $B_{iso}$, low-order $F_g^X$ ($F_{011}$, $F_{002}$), and specimen thickness to minimise the difference between $I_i^{obs}$ (Fig. 2f–h, *obs*.) and $I_i^{cal}$ (Fig. 2f–h, *cal*.) (See "Methods" for details). Both the weighted $R_W$ factor and the standard $R$ factor were minimised. Absolute difference maps between $I_i^{obs}$ and $I_i^{cal}$ are shown as *diff*. in Fig. 2f–h.

For the CBED pattern obtained from the L1$_0$ FePd sample (Fig. 2b), $R_W$ reaches a minimum of 0.1778 at $\eta = 0$ (Fig. 2e). All features and

contours in experimental CBED discs match very well with calculated intensity (Fig. 2f, $\eta = 0$). The rapid increase in $R_W$ with increasing $\eta$ indicates that CBED is sensitive to even slight deviations from perfect order (Fig. 2e and Supplementary Fig. 3). As $\eta$ increases from 0 to 0.5 and 1, the *diff.* maps show more prominent features, indicating growing discrepancies between experimental observations and theoretical calculations (Fig. 2f). By fitting the $R_W$–$\eta$ using a quadratic polynomial (Fig. 2e and Supplementary Fig. 10), the optimised $\eta$ is determined to be 0.855 and 0.856 for the two disordered CBED patterns (Fig. 2c, d), respectively. Surprisingly, the two patterns, despite stemming from regions with substantially different thicknesses (772.2 Å and 4403.4 Å), exhibit similar disorder parameters, indicating a locally uniform ordering throughout the disordered FePd sample. In general, CBED refinement displays substantial sensitive to $\eta$, as indicated by significantly more features in the *diff.* maps for $\eta = 0$ compared to $\eta \geq 0.5$, and higher $R_W$ values for $\eta = 0$ compared to $\eta = 0.86$ (Fig. 2g, h and Supplementary Figs. 4, 5). However, the limited change in the $R_W$ curve beyond $\eta > 0.6$ suggests that the CBED is less sensitive to changes in large disorder as compared to order (Fig. 2e), providing substantial potential for measuring order within disordered regions.

The coupling between $\eta$ and $B_{iso}$ was then evaluated by generating the $R_w$ isosurfaces on a grid of fixed $\eta$ and $B_{iso}$ values ($\eta$ step: 0.1, $B_{iso}$ step: 0.1), while all other parameters were relaxed during QCBED refinement (Fig. 2i–k). A clearly defined global minimum on the $R_w$ isosurface for each of the three CBED patterns verifies simultaneous sensitivity to both $\eta$ and $B_{iso}$. For the ordered sample (Fig. 2i), the global minimum on the grid ($R_w = 0.17783$) is found at $\eta = 0$ and $B_{iso} = 0.2$ Å$^2$, consistent with the refinement result ($\eta = 0$, $R_w = 0.17778$). For the disordered samples (Fig. 2j, k), the grid minima are found at high disorder parameters: $R_w = 0.1407$ at $\eta = 0.9$, $B_{iso} = 0.3$ Å$^2$ for 772.2 Å thickness, and $R_w = 0.1814$ at $\eta = 0.8$, $B_{iso} = 0.4$ Å$^2$ for 4403.4 Å thickness. These grid minima $R_w$ values are slightly higher than the respective refined minima ($R_w = 0.1382$ at $\eta = 0.855$; $R_w = 0.1808$ at $\eta = 0.856$), as expected due to the discrete sampling. The $R_w$ isosurface for the L1$_0$ ordered FePd (Fig. 2i) narrows along the $\eta$-direction, signalling that the refinement is more sensitive to $\eta$ than to $B_{iso}$. In contrast, the disordered sample exhibits an isotropic $R_w$ isosurface, reflecting a comparable sensitivity to both parameters. Similar analyses for $\eta$ - $F_{011}$ and $\eta$ - $F_{002}$ from all patterns (Supplementary Fig. 11) also reveal well-defined global minima, confirming reliable and simultaneous determination of $\eta$, $B_{iso}$, and $F_g^X$ via QCBED refinement.

## CBED refinement for disordered FePd

The highly disordered FCC FePd sample, characterised by large $\eta$ values, served as a model system to study the influence of disorder on low-order $F_g^X$ and DWFs. Given the marginal difference in $R_W$ values for $\eta = 0.86$ and $\eta = 1.0$ (0.1808 versus 0.1816 for the CBED pattern in Fig. 2d), QCBED refinement was simplified by employing a highly symmetric FCC unit cell that assumes a random Fe/Pd distribution (Fig. 3a). This simplification effectively excludes weak ordered reflections, which in turn allows for the use of larger disc radii, capturing a greater number of pixels for more robust refinement. Specifically, this increases the dataset from 44,519 to 147,847 pixels for the CBED pattern in Fig. 2d. Under the MBOZA condition near the FCC [110] zone axis, the (000), (1$\bar{1}\bar{1}$), (002), (2$\bar{2}$0), (1$\bar{1}$3), and (2$\bar{2}$2) discs intersecting the Ewald sphere (Fig. 3b) are used as input for QCBED refinement. $F_g^X$ is defined as the sum of 50% Fe and 50% Pd using CPA, and $B_{iso}$ is assigned as an averaged, isotropic DWF for both Fe and Pd atoms.

In principle, five lowest-order $F_g^X$ ($F_{111}$, $F_{200}$, $F_{220}$, $F_{113}$, and $F_{222}$) contained in Bloch wave calculations near the [110] zone axis can be simultaneously refined to reconstruct $\Delta\rho^{EXP}(r)$. The sensitivity of QCBED refinement to $B_{iso}$ and low-order $F_g^X$ values was assessed by incrementally relaxing parameters (Fig. 3c). For the CBED pattern in Fig. 2d, fixing all $F_g^X$ at IAM values and refining only $B_{iso}$ and thickness yields a high $R_w$ value of 0.2560, with noticeable discrepancies

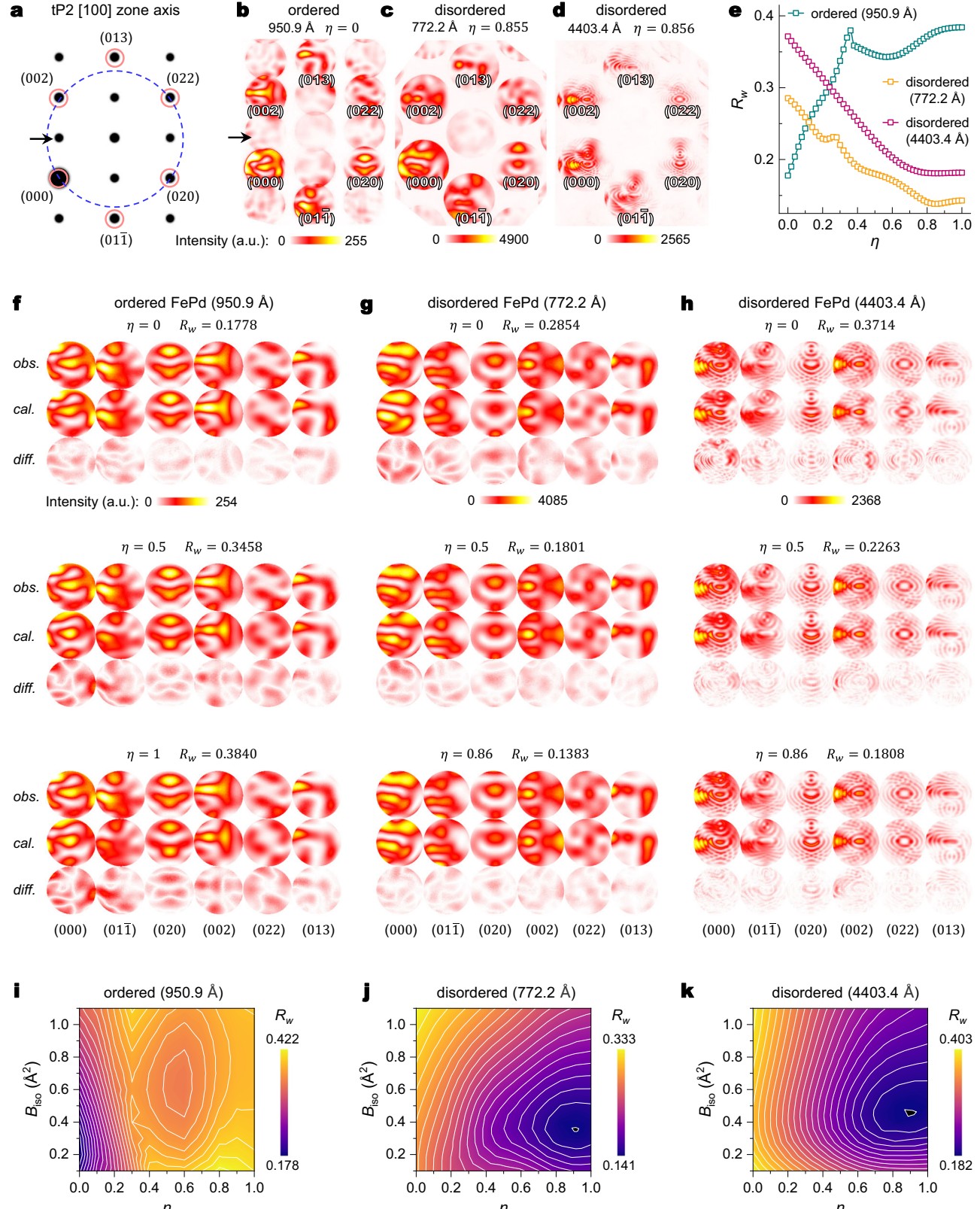

**Fig. 2 | Measurement of the disorder parameter $\eta$ for MBOZA convergent beam electron diffraction (CBED) patterns acquired from ordered and disordered FePd. a** Simulated selected area electron diffraction (SAED) pattern of tP2 FePd along the [100] zone axis. (020), (002), and (022) discs are exactly on the Ewald sphere surface for this MBOZA condition. The blue dashed circle is the intersection between the Ewald sphere and ZOLZ. **b–d** Three experimental MBOZA CBED patterns acquired from ordered (**b**) and disordered (**c, d**) FePd. **e** Weighted R-factor ($R_W$) as a function of $\eta$ for the three CEBD patterns in (**b–d**). **f–h** QCBED refinement results of the three CBED patterns, showing the observed (*obs.*), calculated (*cal.*), and difference (*diff.*, defined as |*obs. - cal.*|) intensity distribution of the six diffraction discs (000), (01Ī), (020), (002), (022) and (013) for different $\eta$. **i–k** $R_w$ isosurface plots calculated from fixed values of $\eta$ and $B_{iso}$ for the three CBED patterns. Source data are provided as a Source Data file.

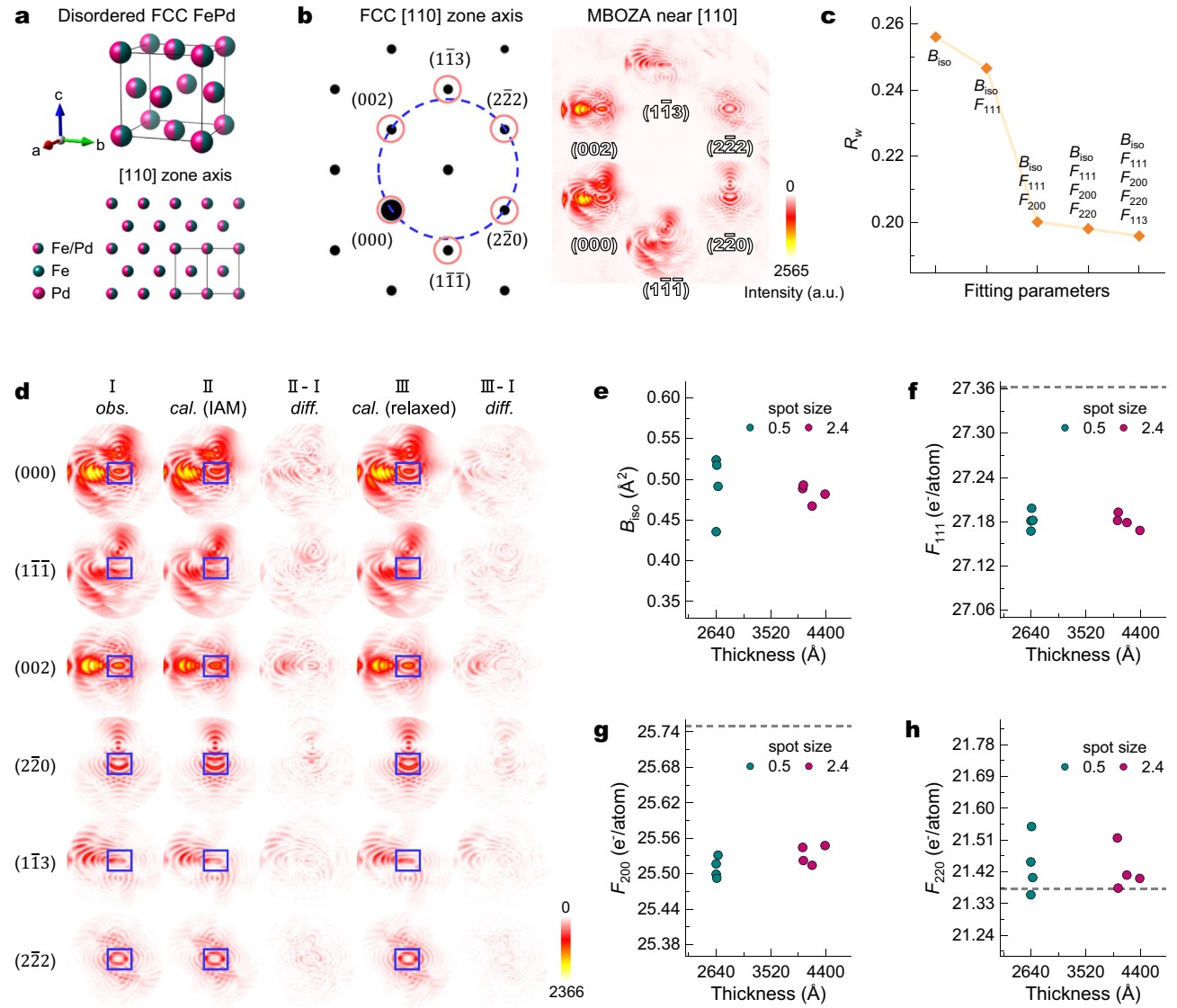

**Fig. 3 | QCBED refinement results of FCC FePd. a** Crystal structure model and projected atomic columns along the [110] zone axis of disordered FCC FePd. **b** Simulation of SAED pattern along the [110] zone axis, and the MBOZA CBED pattern in Fig. 2d re-indexed using FCC unit cell. **c** The change of $R_w$ as more structure factors are included in QCBED refinement. **d** QCBED refinement results showing the observed intensity (*obs.*, I), calculated intensity (*cal.*) with fixed independent atom model (IAM) $F_g^X$ (II), and calculated intensity after relaxing $F_g^X$ (III), for

six diffraction discs (000), (1$\bar{1}\bar{1}$), (002), (2$\bar{2}$0), (1$\bar{1}$3), and (2$\bar{2}$2). The difference maps (*diff.*, defined as |*obs.* - *cal.*|) are shown in (II-I) and (III-I). Blue rectangles denote central regions that are dominated by dynamic diffraction. **e–h** Refinement results as a function of thickness for different probe sizes: $B_{iso}$ (**e**), $F_{111}$ (**f**), $F_{200}$ (**g**), and $F_{220}$ (**h**). The grey dashed lines in (**f–h**) represent the IAM values. Source data are provided as a Source Data file.

between $I_i^{obs}$ (Fig. 3d, column I) and $I_i^{cal}$ (Fig. 3d, column II), primarily in central regions (blue rectangles) dominated by dynamic diffraction. Including $F_{111}$ in the refinement reduces $R_w$ to 0.2465, indicating that QCBED refinement is indeed sensitive to deviations of $F_{111}$ from its IAM value. Further inclusion of $F_{200}$ reduces $R_w$ to 0.2002 (Fig. 3c), approaching the refinement quality observed for L1$_0$ ordered FePd[33] ($R_W$ ~ 0.2) and validating the CPA for disordered FePd. This improvement is visually confirmed by stronger agreement between experimental and calculated disc intensity contours (Fig. 3d, column III). Subsequent inclusion of $F_{220}$ or $F_{113}$ in the refinement yields negligible $R_w$ reduction, suggesting that those structure factors either closely align with IAM predictions or lie beyond the detection limit of this QCBED experiment.

Eight MBOZA CBED patterns acquired at −170 °C were systematically refined using the described incremental approach (Table 1, Fig. 3e–h and Supplementary Tables 1–4). Four of the patterns were acquired from a thin region (~260 nm thickness) with a 0.5 nm probe

size, while the remaining four patterns were acquired from a thicker region (~400 nm) using a 2.4 nm probe size. All patterns were collected with a convergence angle of 5.6 mrad. While inclusion of $F_{220}$ or $F_{113}$ does not significantly improve $R_W$ (Supplementary Fig. 12), it increases estimated errors for $B_{iso}$, $F_{111}$, and $F_{200}$ (Supplementary Fig. 13). Therefore, all values except $F_{220}$ were derived from refinements relaxing $B_{iso}$, $F_{111}$, and $F_{200}$, while $F_{220}$ was obtained by additionally relaxing $F_{220}$ itself. Initial and refined absorption factors are detailed in Supplementary Table 5. Mean refined values from the two distinct probe sizes and thicknesses are in close agreement (Table 1). Notably, the $B_{iso}$ value derived from the thicker region (0.48(1)) has lower estimated uncertainty compared to its thinner-region counterpart (0.49(4)), likely due to the larger probe size and larger thickness capturing a greater variety of local configurations and enhancing statistical consistency (Fig. 3e). Refined structure factors show negligible dependence on probe size (Fig. 3f–h), indicating that $\Delta\rho^{EXP}(r)$ is less sensitive to the local chemical variations as compared to atomic vibrations.

**Table 1 | Refined parameters for eight CBED patterns**

| Image index | Spot size (nm) | Thickness (Å) | $R_w$ | $R$ | $B_{iso}$ (Å$^2$) | $F_{111}$ (e$^-$/atom) | $F_{200}$ (e$^-$/atom) | $F_{220}$ (e$^-$/atom) |
|---|---|---|---|---|---|---|---|---|
| 1 | 0.5 | 2641.1 | 0.1770 | 0.1710 | 0.524 | 27.181 | 25.499 | 21.448 |
| 2 | 0.5 | 2639.9 | 0.2055 | 0.1967 | 0.436 | 27.167 | 25.517 | 21.355 |
| 3 | 0.5 | 2669.4 | 0.1751 | 0.1641 | 0.491 | 27.182 | 25.531 | 21.403 |
| 4 | 0.5 | 2651.2 | 0.2368 | 0.2160 | 0.517 | 27.198 | 25.492 | 21.548 |
| Mean | | | | | 0.49(4) | 27.18(1) | 25.51(2) | 21.44(8) |
| 5 | 2.4 | 4384.9 | 0.2002 | 0.1893 | 0.482 | 27.168 | 25.548 | 21.400 |
| 6 | 2.4 | 4177.8 | 0.1863 | 0.1785 | 0.467 | 27.179 | 25.514 | 21.411 |
| 7 | 2.4 | 4025.8 | 0.2145 | 0.2044 | 0.489 | 27.182 | 25.544 | 21.516 |
| 8 | 2.4 | 4039.4 | 0.1812 | 0.1809 | 0.493 | 27.193 | 25.522 | 21.374 |
| Mean | | | | | 0.48(1) | 27.18(1) | 25.53(2) | 21.43(6) |

## Experimental Debye-Waller factors and low-order structure factors

Comparing ordered and disordered FePd of the same chemical composition provides further insights into the influence of ordering on DWFs and bonding (Fig. 4). Previous work has reported anisotropic DWFs for L1$_0$ FePd at −170 °C as follows[33]: $B_{11}^{Fe} = 0.3$ Å$^2$, $B_{11}^{Pd} = 0.2$ Å$^2$, $B_{33}^{Fe} = 0.2$ Å$^2$, and $B_{33}^{Pd} = 0.3$ Å$^2$. The overall DWF in disordered FePd is substantially larger than those in L1$_0$ ordered FePd at the same temperature. $B_{iso}$ is defined as[41]:

$$B_{iso} = \frac{8\pi^2}{3} \langle u^2 \rangle = \frac{8\pi^2}{3} \left( \langle u^2 \rangle_{dynamic} + \langle u^2 \rangle_{static} \right) \quad (4)$$

where $\langle u^2 \rangle$ is the atomic mean-square displacement (MSD), which is composed of a dynamic component ($\langle u^2 \rangle_{dynamic}$) arising from lattice vibrations and a static component ($\langle u^2 \rangle_{static}$) measuring intrinsic static lattice distortions[42]. The principal difference between disordered and ordered FePd lies in the static component, $B_{static} = \frac{8\pi^2}{3} \langle u^2 \rangle_{static}$, which is zero for perfectly ordered phases but has a non-zero value for disordered FePd due to lattice distortions caused by a random Fe/Pd distribution.

To better understand these static distortions, a $4 \times 4 \times 4$ supercell of FCC FePd containing 256 atoms was constructed using a special quasi-random structure (SQS) method and then relaxed using DFT calculations (Fig. 4a)[43]. The resulting relaxed lattice constants ($a = 15.235$ Å, $b = 15.249$ Å, and $c = 15.238$ Å) demonstrate minimal pairwise deviation (<0.1%), which confirms structural isotropy of the supercell and statistically random Fe-Pd atomic arrangements. Arrows on individual atoms are used to illustrate local lattice distortions in the eight slices of the supercell, which represent displacement from average lattice sites (Fig. 4b). The arrows on Pd atoms are, on average, longer than those on Fe atoms, as confirmed by the displacement amplitude histogram for Pd atoms and Fe atoms (Fig. 4c). The average lattice distortion amplitude for Fe site ($B_{static}^{Fe}$) and Pd site ($B_{static}^{Pd}$) are 0.20 and 0.24 Å$^2$, respectively (Fig. 4d). These calculated static DWF values align well with the observed differences between disordered and ordered FePd, namely $B_{iso} - B_{11}$ and $B_{iso} - B_{33}$ (Fig. 4d), indicating that the increase in DWF is associated with these disorder-generated lattice distortions.

Experimentally-derived low-order structure factors $F_{111}^{EXP}$ and $F_{200}^{EXP}$ are 27.18(1) and 25.52(2) e$^-$/atom, respectively (Table 1 and Fig. 3f, g). Both experimental values are smaller than the IAM predictions, $F_{111}^{IAM} = 27.362$ and $F_{200}^{IAM} = 25.75$ e$^-$/atom. This is consistent with the typical trend that bond formation and charge transfer lead to a smoother electron density distribution and a reduced low-order $F_g^X$[19,34]. For disordered FePd, $\Delta\rho^{EXP}(r)$ was calculated using Eq. (1) incorporating $\Delta F_{111}^{EXP}$ and $\Delta F_{200}^{EXP}$ (Fig. 4f). $F_g^X$s starting from $F_{220}$ are excluded from the $\Delta\rho^{EXP}(r)$ construction because their experimental error bars are significantly larger than those of $F_{111}^{EXP}$ and $F_{200}^{EXP}$

(Supplementary Figs. 13, 14). Consequently, these high-order reflections are indistinguishable from IAM values within the QCBED measurement error (Supplementary Fig. 14 and Supplementary Table 6). This behaviour is consistent with previously observations for other FCC materials such as Al, Ni, and Cu[19,34]. This suggests that $\Delta\rho^{EXP}(r)$ calculated using only $F_{111}^{EXP}$ and $F_{200}^{EXP}$ provides a close representation of the true bonding electron density. Furthermore, a direct comparison of $\Delta\rho^{EXP}(r)$ generated with and without $F_{220}$ confirms that $F_{220}$ has a negligible influence on the result (Supplementary Fig. 15).

The three-dimensional $\Delta\rho^{EXP}(r)$ is visualised using isosurfaces at 0.06 e/Å$^3$ (yellow contour, electron accumulation) and −0.06 e/Å$^3$ (purple contour, electron depletion). For reference, the tetrahedral and octahedral interstitial sites within the FCC unit cell are indicated by grey and yellow spheres, respectively (Fig. 4f). $\Delta\rho^{EXP}(r)$ shows pronounced electron accumulation at the eight tetrahedral sites (black arrows), with negligible electron density at the octahedral sites (Fig. 4f). This behaviour is consistent with previous reports for FCC metals Al[19], Ni[34], Cu[34], and FCC-derived L1$_0$ TiAl[29]. $\Delta\rho^{EXP}(r)$ of L1$_0$ ordered FePd was also calculated using low-order $F_g^X$ (up to $F_{200}$), as previously reported[33] (Fig. 4g). The deviation from IAM for ordered FePd qualitatively matches that of FCC FePd (Fig. 4e), with bonding electrons similarly concentrated at tetrahedral sites (black arrow, Fig. 4g). Strong consistency in $\Delta\rho^{EXP}(r)$ is also observed on the (110) plane between disordered and ordered FePd. Surprisingly, $\Delta\rho^{EXP}(r)$ for FCC FePd using the CPA approach agrees with the ordered FePd, despite that CPA neglects local chemical environment variations in disordered solid solutions. This unexpected agreement suggests that the dominant charge redistribution in FePd arises from global symmetry and coordination effects rather than site-specific chemical disorder.

## Comparison with DFT calculation

To investigate the correlation between $\Delta\rho^{EXP}(r)$ and the local chemical environment in FCC FePd, supercells with varying Fe-Pd nearest neighbour (NN) configurations were designed (Fig. 5). These five supercells all maintain a 1:1 FePd stoichiometry and FCC coordination, ensuring that each atom has 12 NNs. The simplest tP2 supercell, defined using its Pearson symbol[44], replicates the L1$_0$ FePd crystal structure (Fig. 5a), where Fe atoms have 4 Fe and 8 Pd NNs, while Pd atoms have the inverse configuration (8 Fe, 4 Pd NNs) (Fig. 5a). Four larger supercells oP4, tP4, tP8, and oP8 were generated by expanding tP2 supercell along the $a$, $b$, or $c$ axes (Fig. 5b−e). These expansions introduce five additional NN environments for Fe and Pd atoms, with the number of Pd (or Fe) atoms in the NN shell being 4, 6, or 8 (Fig. 5b−e).

The electron density of each supercell was calculated using a full-potential linearised augmented plane-wave (FP-LAPW) method implemented in WIEN2k with the PBE generalised gradient approximation (GGA). The calculated $F_g^X$ corresponding to the (111) and (200) reflections of FCC FePd are listed in Table 2. Additional symmetrically

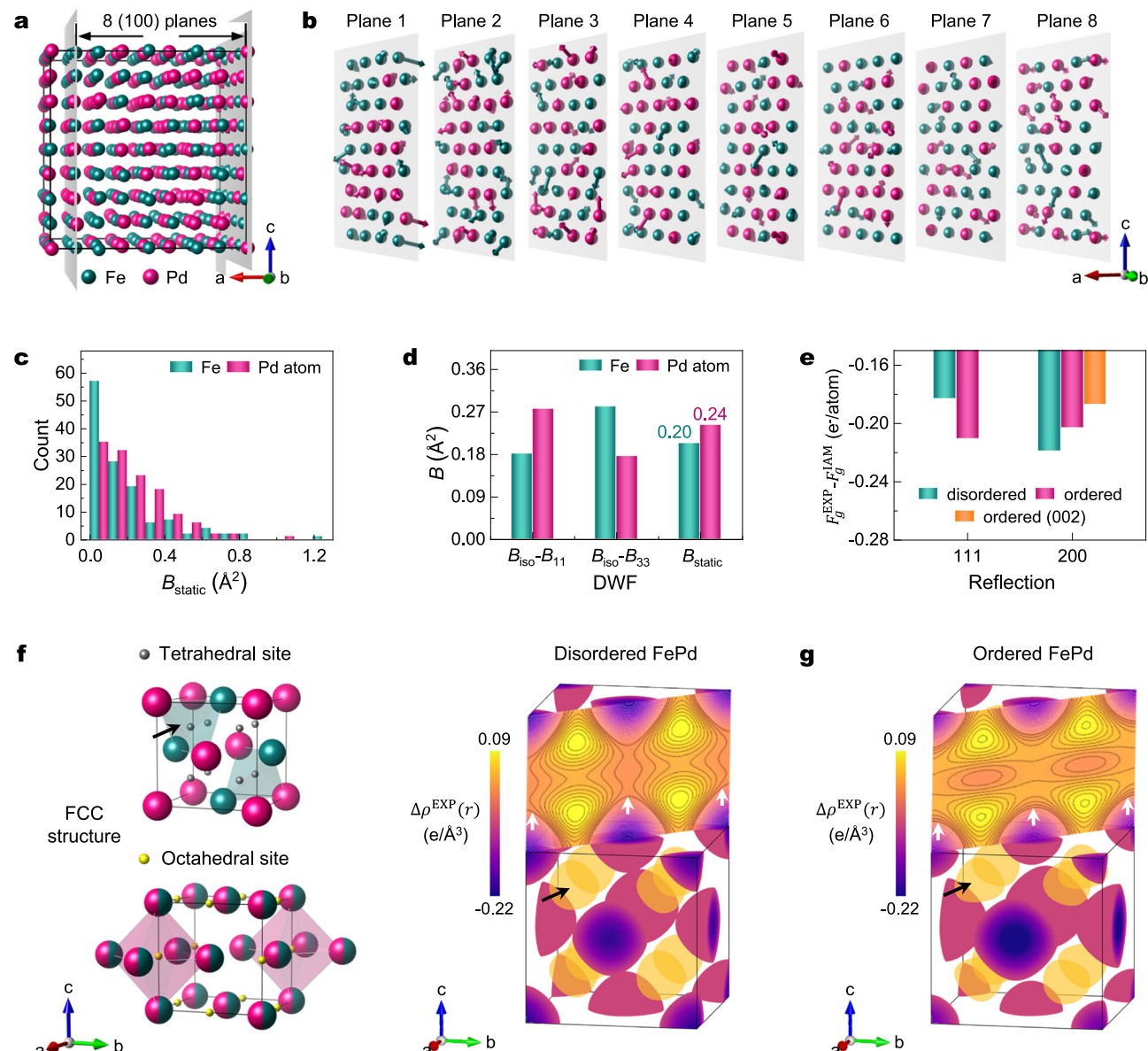

**Fig. 4 | The influence of disorder on Debye-Waller factors (DWF) and deformation electron density map $\Delta\rho^{EXP}(r)$. a** DFT-relaxed supercell structure of disordered FePd. **b** Displacements of Fe and Pd atoms in eight (100) planes from their averaged lattice positions. The arrows are proportional to the displacement vector. **c** Histogram of $B_{static}^{Fe}$ and $B_{static}^{Pd}$ as calculated using the static lattice distortion. **d** Comparison between the difference of ordered and disordered DWF ($B_{iso} - B_{11}$, $B_{iso} - B_{33}$), and the static component $B_{static}$ caused by lattice distortion. **e** Comparison of the difference between experimental and IAM $F_g^X$ for disordered and ordered FePd. **f, g** $\Delta\rho^{EXP}(r)$ of (**f**) disordered and (**g**) ordered FePd constructed

using low-order $F_g^X$s up to $F_{200}$. The upper parts are $\Delta\rho^{EXP}(r)$ on the (110) plane, where the first black line (white arrows) has zero electron density. Black lines within yellow-orange regions and white lines within deep purple-magenta regions represent positive and negative electron density, respectively. The difference between two adjacent contours is 0.01 e/Å³. The tetrahedral and octahedral interstitial sites within an FCC unit cell are marked with grey and yellow spheres, respectively. The tetrahedral sites of electron accumulation are indicated by black arrows. Source data are provided as a Source Data file.

distinct $F_g^X$ values up to $\frac{\sin\theta}{\lambda} = 0.263$ Å⁻¹ are provided in Supplementary Tables 7–11. Remarkably, the $F_{111}$ and $F_{200}$ values across all supercells agree with each other within 0.01 e⁻/atom, despite the significant variations in their NN configurations. The calculated $F_{111}$ values are systematically smaller than the experimentally observed $F_{111}^{EXP}$, while the calculated $F_{200}$ values exceed $F_{200}^{EXP}$. These discrepancies between the calculated (DFT) and experimental (CBED) structure factors for $F_{111}$ and $F_{200}$ may arise from interactions involving the $d$-electrons of Fe and Pd, as suggested by prior studies[34].

Deformation electron density $\Delta\rho^{DFT}(r)$ was then generated for each supercell (Fig. 5a–e) by incorporating low-order structure factors up to $\frac{\sin\theta}{\lambda} = 0.263$ Å⁻¹, corresponding to the (200) reflection of FCC

FePd. A striking contrast emerges when comparing $\Delta\rho^{DFT}(r)$ with $\Delta\rho^{EXP}(r)$ (e.g., Fig. 4f): all supercells exhibit pronounced electron accumulations at octahedral interstitial sites, as indicated by black arrows and highlighted by pink polyhedrons (Fig. 5a–e, octahedral sites). In FCC-based crystal lattices, octahedral sites are coordinated by six neighbouring atoms. Notably, electron accumulation at these sites occurs exclusively when at least four of the surrounding atoms are Pd (Fig. 5a–e). The prevalence of octahedral sites displaying such electron enrichment diminishes progressively with increasing supercell size (Fig. 5f). It is expected that beyond a critical supercell dimension, electron accumulation will shift entirely to tetrahedral interstitial sites, a trend consistent with CBED observations.

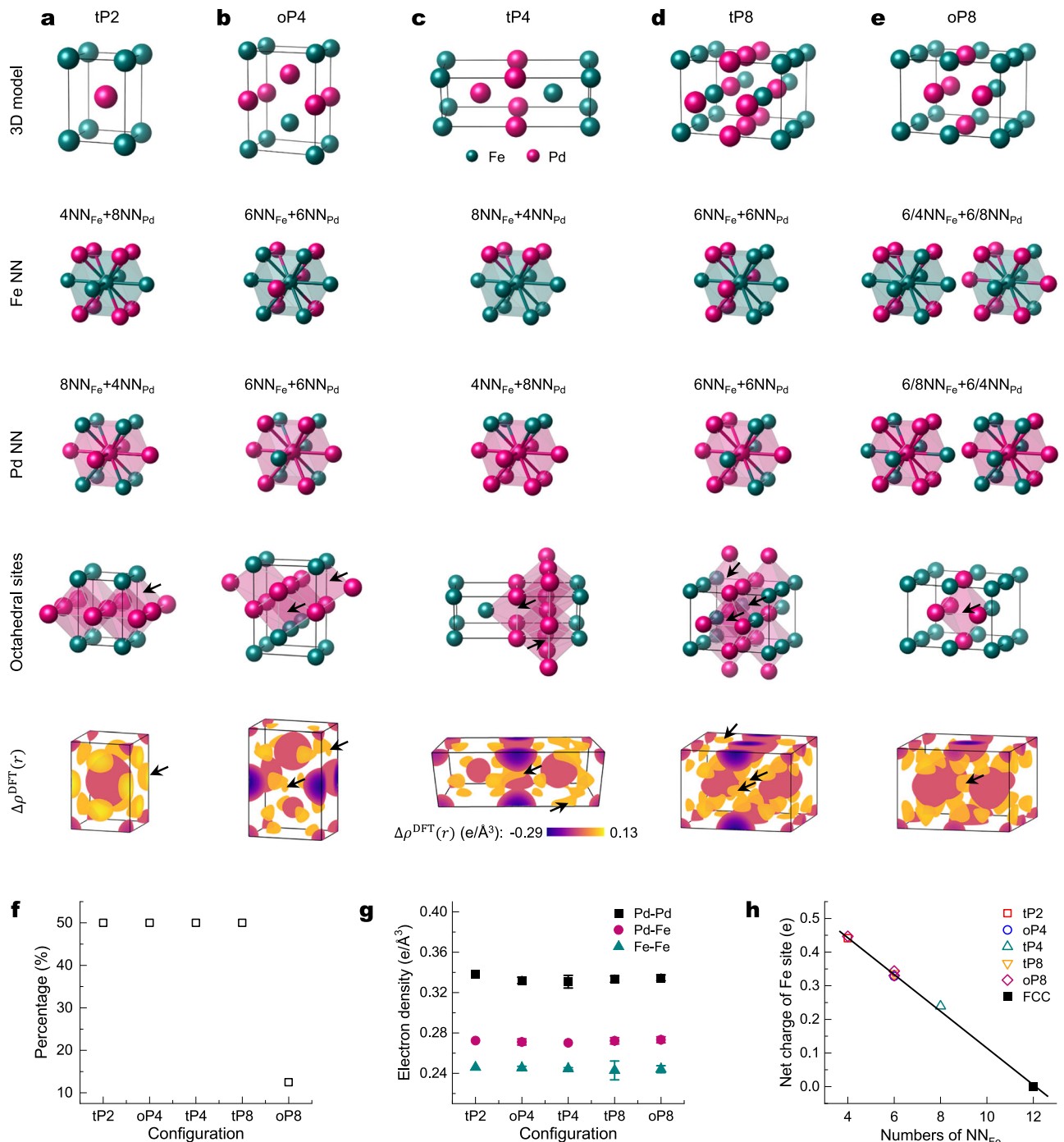

**Fig. 5 | The influence of the local chemical environment on bonding in disordered FCC FePd. a–e** Crystal structure model, the coordination of Fe nearest neighbour (Fe NN) and Pd nearest neighbour (Pd NN) atoms, octahedral sites (pink sites represent the positions of electron accumulation), and $\Delta\rho^{DFT}(r)$ of five different structures: (**a**) tP2, (**b**) oP4, (**c**) tP4, (**d**) tP8, and (**e**) oP8. Electron accumulation at octahedral sites is indicated by the black arrows. **f** Percentages of octahedral sites with electron accumulation for different supercells. **g** Bonding strength between the nearest neighbours for different atom pairs (Pd-Pd: black, Pd-Fe: pink, Fe-Fe: teal). The error bars represent the standard deviation calculated from independent measurements (see Supplementary Table 12) for data points. **h** The charge of an Fe atom as a function of the number of $NN_{Fe}$ atoms. The black line represents a linear fit to the data, with the net charge of the Fe site fit by $-0.055$ $NN_{Fe} + 0.666$. Source data are provided as a Source Data file.

Analysis of maximal electron density across supercells (Fig. 5g) reveals a bonding strength trend of Pd-Pd > Pd-Fe > Fe-Fe, where Pd-Pd bonds were the strongest and Fe-Fe the weakest. This trend supports the thermodynamic hypothesis that binary AB random solid solutions can be described by A-A, B-B, and A-B bond statistics[45,46], which can be used to predict thermodynamic properties[47,48]. The net charge of Fe atoms in disordered supercells was determined by analysing the full electron density generated by WIEN2k using the atom in molecules (AIM) method (Fig. 5h)[49]. Consistent with the Pauling electronegativities of Fe (1.8) and Pd (2.1), a net positive charge was observed on Fe, indicating electron transfer from Fe to Pd[50]. A linear correlation was found between the net charge of an Fe atom and the number of its NN Fe atoms (Fig. 5h). Extrapolation of this linear relationship to the case of an Fe atom

**Table 2 | $F_{111}$ and $F_{200}$ from different supercells using DFT**

| Supercell | tP2 | oP4 | tP4 | tP8 | oP8 | CBED | IAM |
|---|---|---|---|---|---|---|---|
| $F_{111}$ (e⁻/atom) | 27.146 | 27.151 | 27.149 | 27.148 | 27.148 | 27.18(1) | 27.362 |
| $F_{200}$ (e⁻/atom) | 25.575 | 25.575 | 25.578 | 25.576 | 25.572 | 25.52(2) | 25.750 |

The CBED and IAM results are also listed here for comparison.

surrounded by 12 NN Fe atoms predicts a vanishing net charge, consistent with charge neutrality in pure metallic FCC Fe.

In summary, the disorder parameter, low-order $F_g^X$, and DWF of disordered FePd were simultaneously measured using MBOZA QCBED method combined with CPA. By optimising the processing of experimental CBED intensity data and the QCBED refinement procedure, disordered FePd achieves refinement results comparable in quality to those of ordered FePd. Comparing disordered and ordered FePd reveals that disorder significantly increases the static component of DWF but has minimal influence on low-order $F_g^X$. DFT simulations using supercells accurately predict the low-order $F_g^X$ and provide detailed insights into charge transfer between Fe and Pd as a function of the local chemical environment. These findings demonstrate the potential of combining QCBED and DFT for investigating $\Delta\rho^{EXP}(r)$ in disordered materials.

## Methods

### Sample preparation

A nominally equiatomic FePd alloy was synthesised from high-purity Pd (99.95%) and Fe (99.98%) by vacuum arc melting under purified argon atmosphere, followed by casting into a water-cooled copper hearth. Rectangular sections (~50 mm long, 12 mm × 6 mm cross-section) were cut from the as-cast alloy and subjected to a 50% thickness reduction via cold rolling. The rolled specimens were then recrystallised at 950 °C for 6 h under a residual atmosphere of purified argon gas followed by quenching into ice-brine. A cylindrical single crystal (~40 mm length, 12 mm diameter) of the disordered FCC phase was then grown from the homogenised material using the Bridgman technique. For microstructural analysis, 3-mm discs were punched from the plates, mechanically thinned to approximately 50 μm, and electropolished at 4 °C using an electrolyte consisting of 82 vol% acetic acid, 9 vol% perchloric acid, and 9 vol% ethanol.

### CBED pattern acquisition

Zero-loss-energy-filtered CBED patterns were acquired using a JEOL JEM2100F TEM equipped with a post-column energy filter (Gatan Tridiem). An energy-selecting slit with a width of 10–15 eV was used. A double-tilt low-background cold stage was used to cool the TEM specimens to temperatures as low as liquid nitrogen, ~96 K (−177 °C). A focused electron probe with a diameter of 0.5 nm or 2.4 nm was employed. CBED patterns were recorded on a charge-coupled device (CCD) camera with a maximum resolution of (2048 × 2048) pixels.

### DFT simulation

Special quasi-random structure (SQS) of disordered FePd was generated using a Monte Carlo algorithm[51] implemented within the Alloy Theoretic Automated Toolkit (ATAT)[52]. A 4 × 4 × 4 FCC supercell containing 128 Fe, 128 Pd atoms was used to model the disordered structure. The SQS includes 6 types of 5th-order clusters with diameters equal to the FCC unit cell lattice constant. Structure relaxation was performed using the Vienna ab initio simulation package (VASP)[53] with the Perdew-Burke-Ernzerhof (PBE) functional within the generalised gradient approximation (GGA)[54]. A kinetic energy cutoff of 450 eV and a 2 × 2 × 2 Monkhorst-Pack k-point mesh were used[55,56]. The lattice parameters $a$, $b$, and $c$ were allowed to relax, while the three angles $\alpha$, $\beta$, and $\gamma$ were fixed at 90°. Atomic positions were optimised using the

conjugate gradient method until the total energy change was less than $10^{-6}$ eV and the force on each atom was less than 0.05 eV/Å.

Theoretical $F_g^X$s were calculated using the WIEN2k package based on the LAPW + local orbitals (lo) method[57–59]. The exchange-correlation potential $V_{xc}$ was approximated using the PBE version of GGA. A k-point mesh of 10,000 points within the unit cell and a $R_{MT}*k_{max}$ value of 10 were used, ensuring convergence of the calculations. Spin-orbital coupling was not included.

### CBED refinement

QCBED refinements were performed using the MBFIT software[23]. Experimental intensities ($I_i^{obs}$) were extracted from CBED discs on the Ewald sphere, while theoretical intensities ($I_i^{cal}$) were simulated using Bloch wave theory[20,22,39,60], employing an average potential for FCC FePd. DWFs and $F_g$ are simultaneously refined[29,33]. The QCBED refinement minimises the following function, $T$:

$$T = \sum_i \frac{\left(I_i^{obs} - cI_i^{cal}\right)^2}{\sigma_i^2} \tag{5}$$

using the Levenberg-Marquardt algorithm. Here, $\sigma_i$ represents the intensity uncertainty in the $i^{th}$ pixel and is defined as $\sigma_i^2 = \sigma_{ip}^2 + \sigma_{bkgd}^2 + \Delta(I_i^{obs})^2$, where $\sigma_{ip} = \sqrt{I_i^{obs}}$, $\sigma_{bkgd}$ is the constant background intensity, and $\Delta$ represents linear gain noise[61,62]. Following a sensitivity analysis (Supplementary Fig. 16), the contribution from linear gain noise was found to have a negligible impact on the refined parameters and conclusions for this study and was therefore not explicitly included in refinement. $c$ is a scaling factor[63]. The weighted R-factor ($R_W$) and the standard R-factor ($R$) are defined as:

$$R_W = \left( \frac{\sum_i \frac{(I_i^{obs} - cI_i^{cal})^2}{\sigma_i^2}}{\sum_i \frac{(I_i^{obs})^2}{\sigma_i^2}} \right)^{\frac{1}{2}} \tag{6}$$

$$R = \frac{\sum_i \left| I_i^{obs} - cI_i^{cal} \right|}{\sum_i I_i^{obs}} \tag{7}$$

Smaller $R_W$ and $R$ values indicate a better agreement between $I_i^{cal}$ and $I_i^{obs}$. The uncertainty in the refined parameters is propagated from $\sigma_{ip}$ through the Jacobian as described in ref. 22.

The initial values for the real part of electron structure factors ($U_g$) were converted from IAM X-ray structure factors $F_g$ using Mott formula[37]. The imaginary part of $U_g$, representing absorption, was calculated using the method described by Bird and King[40]. During refinement, the sample thickness, two or three low-order $U_g$ values including the absorption factors, and the DWFs, were independently relaxed. In each refinement step, the high-order $U_g$ values were updated using the DWFs from the previous step and IAM $F_g$ values. Several hundred $U_g$ values were included in the Bloch wave simulation. After reaching the global minimum, low-order $F_g$ values were obtained from the optimised low-order $U_g$ and DWFs using the Mott formula.

For the disordered solid solution, an average potential $U_g^{\mathrm{avg}}$ was used in Bloch wave formalism:

$$U_g^{\mathrm{avg}} = \sum_i a_i U_g^i \qquad (8)$$

where $a_i$ represents the atomic percent of element $i$ in the solution, and $U_g^i$ is the electron structure factor assuming the solid solution is fully occupied by element $i$. For the equiatomic FCC FePd, this expression simplifies to:

$$U_g^{\mathrm{avg}} = \frac{U_g^{\mathrm{Fe}} + U_g^{\mathrm{Pd}}}{2} \qquad (9)$$

The absorption, i.e., the imaginary part of $U_g^i$, was also treated similarly. An average isotropic DWF ($B_{\mathrm{iso}}$) was assigned to both Fe and Pd atoms, an approximation justified by the similar DWFs observed for Fe and Pd in L1$_0$ FePd. After a simultaneous refinement of low-order $U_g^{\mathrm{avg}}$ and $B_{\mathrm{iso}}$, the average X-ray $F_g^{\mathrm{avg}}$ can be calculated using the Mott formula:

$$F_g^{\mathrm{avg}} = 2(Z_{\mathrm{Fe}} + Z_{\mathrm{Pd}})e^{-B_{\mathrm{iso}}s^2} - \left(\frac{C\Omega s^2}{\gamma}\right)U_g^{\mathrm{avg}} \qquad (10)$$

where $s = \frac{|g|}{2}$, $\Omega$ = volume of the unit cell, $\gamma$ = relativistic constant, $C = 131.2625\text{Å}$, and $Z_{\mathrm{Fe}}$ and $Z_{\mathrm{Pd}}$ are the atomic numbers for Fe (26) and Pd (46), respectively.

## Data availability
The data that support the findings of the study are included in the main text and supplementary information files, with comprehensive datasets available in the Source Data files. Source data are provided with this paper.

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

## Acknowledgements

This work was supported by the National Natural Science Foundation of China (12474020).

## Author contributions

J.W. (Jörg Wiezorek) and X.S. conceived and supervised the research. X.S., W.L., Z.X., and A.K. acquired CBED patterns and conducted QCBED refinements. X.S., W.L., W.C., H.R., and W.Z. performed the DFT simulations and associated analysis. W.L., Z.X., G.V.T., J.W., J.S.W. (Jinsong Wu), and X.S. led the manuscript draughting. All authors contributed to data interpretation, participated in discussions, and reviewed/edited the manuscript.

## Competing interests

The authors declare no competing interests.
