## [Transparent Peer Review file · Nature Communications]

Revealing disorder parameter and deformation electron density using electron diffraction

Corresponding Author: Professor Xiahan Sang

Version 0:

Reviewer comments:

Reviewer #1

(Remarks to the Author)

Comments on "Revealing disorder parameter and bonding electron density using electron diffraction" by Lin et al. NCOMMS-24-59352

Summary:

The work presented introduces a disorder parameter, η , into the IAM structure factor calculation that is used to generate the reference structure factors in Bloch-wave calculations of CBED patterns being matched to experimental ones in a QCBED analysis of FePd. The bonding electron density is determined from the refined lower order structure factors and their differences from the IAM structure factors which incorporate the QCBED-optimised value of η .

The bonding electron densities obtained from QCBED-measured structure factors represent the average structure in the experimentally probed nanoscale volumes of disordered FePd and these are compared with DFT simulations.

This is a very interesting piece of work and a commendable direction for new research in QCBED, but there are many major scientific flaws that stem largely from an incomplete treatment of the problem and a lack of detail when it comes to treating systematic errors that are embedded in CBED data. The most fundamental error though, is one where the evidence that the authors have in front of them contradicts their approach to the problem. They have themselves concluded that Fe and Pd have different electronegativities and therefore, the approximation that FePd behaves as if it were an elemental metal in terms of bonding (allowing all structure factors other than F111 and F200 to be neglected in a bonding charge density analysis), is fundamentally wrong. See points N and onwards below.

The issues with this work and its presentation are numerous and significant and, in some cases, severe. I will outline these below in terms of issues with the manuscript and the problems with the science. The work requires major revision, but I do believe that if these revisions are carried out, it is certainly worthy of publication in Nature Communications and I support the authors wholeheartedly in their efforts to publish in this esteemed journal.

The manuscript:

1. Page 1: I noticed that the author, Van Tendeloo is misspelled in the author list of the manuscript on page 1. This is very careless and is symptomatic of the many errors in the writing of the manuscript.

I advise the author (and the co-authors) to carefully proof-read manuscripts BEFORE submission. It is a pity that referees have to navigate so many mistakes which act as a fog in assessing work that does actually have potential.

2. Page 2: "using the simultaneously imaging mode" on page 2 is poor English and incorrect grammar. There are many similar cases everywhere in the manuscript and I suggest that the authors engage the help of someone who writes English well to check their document. I am surprised that some of the co-authors did not pick these very obvious mistakes up in proof-reading.

3. Page 2: Two-beam is referred to incorrectly here. Very few true 2-beam studies have ever been published. The authors mean systematic-row QCBED, not 2-beam QCBED.

4. Page 2: ρ^{EXP} : The "EXP" is not defined anywhere - do the authors mean experimental? If so, then equation 1 is not strictly an experimental total electron density because QCBED cannot measure all of the structure factors. It is possible to produce an experimental deformation electron density, $\Delta\rho^{\text{EXP}}(r)$, where QCBED measures all of the bonding-affected structure factors (and thus all the IAM structure factors cancel out in producing the difference, but it is not possible for QCBED to determine a total bonding electron density as it would have to measure all structure factors to high $\sin\theta/\lambda$).
5. End of page 2: "via Fourier transform" is poor English.
6. Page 3, line 65: "has" should be "have" as this is currently a singular/plural contradiction.
7. Top of page 3: The first sentence is wrong. As said before, QCBED cannot measure $\rho^{\text{EXP}}(r)$. It has been used to determine $\Delta\rho^{\text{EXP}}(r)$ and the cited publications bear this out.
8. Page 4, line 102: "MBOZA condition means..." is a bad way to start a sentence. This mode of expression is poor English.
9. Page 4, lines 107-108: "whether the MBOZA condition is also sensitivity to change..." – again poor English. You mean "sensitive".
10. Page 4: I am certain that η is missing somewhere in equation 2. It is in equation 3 and looking at the dependence of F_g ($h + k + l$ even) on η in Fig. 1c, it is obvious that η has been left out of equation 2. This is extremely careless and I am astonished that this was missed in proof-reading the manuscript.

11. Page 4: Supplementary Fig. 1 is totally superfluous. All one ever needs to say is where the scattering factors came from. Your audience does not need the obvious shown to them - they all know what scattering factors look like as a function of s .

Furthermore, when citing from where the scattering factors came from, reference 36 is far too general and not specific enough. You need to specify that they are in volume C of the International Tables for Crystallography. Furthermore, at this point, you should actually cite Doyle and Turner (ref. 38) first as they originated these scattering factors. Reference 36 should be mentioned as a secondary reference.

12. References section: There are few duplicated references. Please check your references section for duplicates.

13. Page 4, lines 114-116: "...plotting the difference $\Delta F_g X$..." is a tautology because $\Delta F_g X$ is already a difference. If you want to say the word difference, you would say "...plotting the difference in structure factors, $\Delta F_g X$,..."

You are claiming that the difference being plotted is between $F_g X(\text{Biso}, \eta, s)$ and $F_g X(0, 1, s)$ but this is an incomplete description because the plot in Fig. 1c shows only cases where $\text{Biso} = 0$. Please make your description more accurate.

14. Fig. 1 caption: s is not the scattering vector!! It is $|g|/2$. s is a scalar, NOT a vector. This manuscript is very poorly written.

15. Fig.1 caption: The last part of the last sentence should be rewritten as it contains a singular/plural contradiction.

16. Page 5, line 136: Missing "the" before "Ewald sphere".

FROM HERE ONWARDS, I WILL NO LONGER HELP WITH CORRECTING BASIC ENGLISH AND GRAMMAR - PLEASE ENGAGE SOMEONE TO HELP TIDY UP YOUR ENTIRE MANUSCRIPT.

17. Page 5, lines 140-143: The sentence "The (0kl)-type disks with odd $k + l$... or 'disordered' CBED pattern (Fig. 2d)." requires an explanation. It is simply not appropriate to make the observation of absent disks without trying to furnish the observation with any kind of explanation.

18. Throughout the manuscript, the authors label structure factors as F_{ghkl} . This is incorrect. Given that g is defined by hkl , then it suffices to write F_{hkl} and it is incorrect to write it otherwise as to do so would be a symbolic tautology.

19. Caption to Fig. 3: "a completed disordered MBOZA CBED pattern." This implies that the CBED pattern is disordered!! Furthermore, the grammar is incorrect - I think you mean "completely disordered". It should be rewritten as "an MBOZA CBED pattern from a completely disordered region of the specimen."

20. Page 10, line 248: What do you mean by saying that the CBED patterns were statistically analysed? Don't you mean that the results from QCBED of the 8 patterns were statistically analysed?

21. Page 10, line 254: You should quote the results in the reverse order because the sentence leads one to believe that you are assigning the first result in parentheses to the larger probe size (mentioned most recently). This way of quoting results is confusing and reflects poor writing.

22. Page 10, line 255: "more different" is a poor conjunction in English. Please re- write this part.

23. Page 11, line 278: Within 0.1% error of what reference? If one claims an error, one must state unequivocally what the

error is relative to.

24. Only in the caption to Fig. 4 is DFT mentioned for the first time. This should be fully explained in the text as per point L below.

25. Fig. 4, page 12 & line 320 on page 13: Why are only some of the tetrahedral interstices and only some of the octahedral interstices shaded in figure 4f?

26. Page 13, line 316: The difference structure factors are missing units.

Scientific issues:

A. "For $\eta = 0$, the structure is fully chemically ordered, while $\eta = 1$ indicates complete chemical disorder." What if $\text{Occ}(\text{FePd}) = 0.5$ in a manner where a perfectly ordered superlattice is formed? In that case, the "disorder" parameter $\eta = 1$ but you could still have a perfectly ordered and periodic superlattice. This means that your definition of the disorder parameter is not canonical.

Furthermore, you would see the presence of a superlattice in your diffraction patterns. Do you see any extra reflections in some of your CBED patterns?

B. Figure 1: The drawing in part b is incorrect. The transmitted beam (or 000) is always in the ZOLZ, never the FOLZ. The Laue circle of Bragg-satisfied reflections that includes the central beam, 000, is always part of the ZOLZ by definition.

THE ERRONEOUS USE OF FOLZ INSTEAD OF ZOLZ OCCURS IN MANY OTHER PLACES IN THE MANUSCRIPT. PLEASE FIX THIS.

C. Figure 2: In the pattern from the disordered material, there appears to be no noticeable disk perimeter. This may be symptomatic of the CBED probe averaging over a range of varying lattice parameters. What has been done with the QCBED refinement code, MBFIT, to accommodate variations in, or averaging over, fluctuating lattice parameters?

This constitutes a major systematic error that must somehow be accounted for within the refinement algorithm and/or processes.

D. After line 146 on page 5, it becomes clear that η is not a freely refineable parameter within the QCBED software that has been applied, i.e. MBFIT. This means that the authors have changed η and re-refined for fixed values of this parameter. While this affords the reduction of parameter correlations with respect to this parameter, it does not allow it to be fine-tuned in a continuous fashion. This is a weakness that needs to be overcome while at the same time, developing a refinement scheme that subdues parameter correlations.

E. Very few CBED patterns are actually refined in this work which does not allow for an assessment of the reproducibility of the results.

This work would be more significant if the authors were to produce a map of disorder by measuring η from say 400 CBED patterns collected in a grid of 20 x 20 points spanning the spatial range of order fluctuations within the specimen.

F. The R_w results do not change much with increasing η above about 0.6 as the authors correctly state. In QCBED, geometric distortions in the data need to be fully accounted for because these have a much larger effect on R_w than the small changes due to η .

In the absence of any discussion at all on the effects of geometric distortions on the QCBED results, it is highly doubtful that there is much weight in the statement "CBED refinement is generally sensitive to η ".

Further on this same point, the Methods section gives no information as to how any of the systematic and random errors in the CBED data were dealt with. For example:

- How was the point spread function measured and removed?
- How was the noise as a function of signal determined? Note that most detectors are never Poisson so to assume that $\sigma_i \propto \text{liobs}$ in equation 6 in the Methods would be very wrong. This alone can result in refined parameters in shallow parts of the R_w surface in the N-dimensional parameter space of a QCBED refinement, being in greater error than the differences the authors claim to be resolvable when it comes to determining the optimal η from the more disordered regions.

Continuing on point F, the authors later on page 6 state that the "CPA does not fully capture...". While these statements are true, the authors may not be getting the most out of their application of the CPA because they have compromised the potential of what they have set out to do by not dealing with all aspects of the data collection, processing and refinement as accurately as possible.

In the above light, to claim "the disorder parameter of the three regions was successfully measured using CBED...", at the bottom of page 6, is highly dubious.

G. If F220 was refined as stated on page 8, line 213, then why are only graphs of refined F111 and F200 versus thickness shown in Fig. 3 (e & f)? For completeness, the graph of refined F220 versus thickness should also be shown.

H. At the bottom of page 9 and over onto page 10, the following statement is made: "A better agreement, achieved by relaxing the low order Fg in the first refinement, verifies that the low order Fg contribution encoded in the CBED pattern can be extracted using MBOZA CBED technique and the CPA approach for a disordered structure."

This statement is wrong because all that has been shown is that there is an improvement in fit compared to the fit between the IAM pattern and the experimental one. This will always be true regardless of the veracity of the parameters that have been adjusted. You will always get a better fit than for the IAM, for any parameter you switch on in addition to thickness because the freely adjustable parameter will simply act as a buffer.

You need to develop a very solid refinement strategy that reduces parameter correlation and returns meaningfully interpretable parameters. The statement quoted above is a statement of the obvious and does not lead to anything beyond the obvious. It also does not allow much meaning to be drawn from the refined values of the parameters because the manner of the refinement is absent from such a statement.

I. Does beam damage play a role in the collection and analysis of the data? The authors should give the threshold energies for ballistic damage for both iron, palladium and the ordered intermetallic FePd. The latter can be estimated if no specific measurements exist in the literature.

J. Page 10, lines 257-259. "...indicating that the local bonding charge density is not as sensitive to the local chemical environment compared to atomic vibrations" is a very poor conclusion which does not make physical sense. One can only make such statements if one has subtracted the IAM structure factor from the measured structure factors, then the differences between the two sets of measurements will be put in the context of bonding electron (or difference) structure factors and not total electron density structure factors. You cannot make claims about bonding if you have not removed the IAM from your structure factors.

K. Table 1, Page 10: What is the uncertainty associated with all the individual measurements? It is not sufficient to provide measurements from each CBED pattern without gauging the uncertainties of the individual measurements. There are many different ways to do this, but it is not clear that authors have determined local (individual) uncertainties.

L. The authors discuss atomic displacements and local lattice distortions in a quasi-random 4x4x4 super cell on Page 11, lines 276-288. It is not clear from this paragraph as to how the lattice parameters were relaxed or the atomic positions were relaxed. No mention has been made of DFT so far or in this paragraph. See point 24.

M. Page 11, line 291: You state that Fg starting from 220 onwards are not included in the refinement but this contradicts the statement that you did refine F220 on page 8, line 213. This relates to an earlier question as to why the refined values of F220 were not plotted in Fig. 3. There is a reference in the text on page 8 to two refinements being performed consecutively and the first is discussed, involving F220 but I cannot see any clear reference to a discussion of the second refinement. If there is a lot of material discussed between the first and second refinements, then the authors should rewrite the text so that it is clear as to which one is being discussed and attention should be drawn back to the refinement overview (2 refinements) when commencing discussion of the second refinement if this is indeed what is being discussed on page 11. This needs to be made a lot clearer.

N. Further to the leaving out of F220 in the refinements discussed on p11, lines 291- 293, to assume that it is reasonable to leave this structure factor out of refinements because it has assumed the IAM value in the cases of pure elemental FCC metals is naïve to say the least. The dissimilar electronegativities of Fe and Pd would make it unlikely that bonding is as delocalised as in pure elemental metals, and therefore, the assumption that F220 will assume the IAM value is highly questionable.

Where is the evidence that the authors can present for FePd to justify this approach?

THE BEST TEST FOR THIS IS TO COMPUTE STRUCTURE FACTORS USING DFT FOR FePd AND COMPARE THEM TO THE IAM. THIS WILL GIVE SOME SORT OF AN ESTIMATE AS TO HOW MANY Fhkl ARE AFFECTED BY BONDING. RECENT STUDIES SHOW THAT IN INTERMETALLICS, 20-25 STRUCTURE FACTORS (SYMMETRY UNIQUE) ARE BONDING AFFECTED. TRUNCATION TO THE LOWEST 2 STRUCTURE FACTORS IN $\sin\theta/\lambda$ IN THIS WORK IS ALMOST CERTAINLY NOT GOING TO GIVE AN ACCURATE ASSESSMENT OF BONDING IN AN INTERMETALLIC. LOOKING AHEAD TO THE END OF THE PAPER, IT SEEMS THAT THE MAPS FROM DFT ALSO NEGLECT HIGHER ORDER STRUCTURE FACTORS BEYOND F200. TO COMPUTE SUCH A HIGHLY TRUNCATED FOURIER SERIES WHEN SO MANY MORE TERMS ARE SURELY GOING TO BE NON-ZERO IS A MAJOR FLAW.

O. Page 11, line 295: What do the authors mean by "flatten the electron density". This needs to be explained carefully.

P. Fig. 4, page 12: The deformation electron density plots are almost certainly incorrect because it is highly likely that bonding-affected structure factors have been ignored. Refer to point N above.

Q. Page 13: All discussions of bonding electron densities relating to Fig. 4 are invalid as the authors have probably ignored

a large number of bonding-affected structure factors. See point N above.

R. Section 2.4: The whole section neglects any mention of how the bonding electron density was determined from DFT. This leaves one to assume that the same method as in the previous section was applied which was to only consider F111 and F200 and their symmetry equivalents. As per point N above, this approach is wrong.

Continuing with this point and point N, even table 2 suggests that only refining F111 and F200 is a flawed approach because whilst there is little difference between the QCBED and DFT results for F111, there is actually a statistically significant difference between the QCBED and DFT results for F200, implying that there is some compensation occurring due to the neglect of higher order structure factors deviating from the IAM.

At the bottom of page 14, the use of the AIM method implies that the total charge density (all structure factors) is being used in the net charge analysis. This would contradict the treatment used so far which is to only consider F111 and F200. Furthermore, the AIM net charge analysis is explained by the differences in electronegativities between Fe and Pd, which is precisely the point I made in point N above. So, the authors own conclusions contradict the exclusion of all structure factors apart from F111 and F200 and their symmetry equivalents. Is this not an obvious self-contradiction of this work?

S. Lines 376 and 377: The authors tie up their explanation with the argument of elemental FCC Fe. But iron in its elemental ground state is BCC. A DFT calculation will verify this. This is also a significant self-contradiction.

In conclusion, I think this work has great potential but all of the fundamental flaws discussed above need to be addressed and fixed before I can recommend acceptance. This should not be too difficult for the authors to do – all they need to remember is to make their analysis physically sensible on all fronts and provide all the necessary evidence to support the approximations made. Some approximations that have been made are clearly self-contradictory and ignore the evidence right in front of the authors.

I am happy to referee this work again until it is acceptable for publication.

Reviewer #2

(Remarks to the Author)

This paper by Lin et al. addresses a very timely problem, that is the degree of disorder and impact on bonding in intermetallic alloys, which is very much relevant to recent interest in complex alloys, including high-entropy alloys. The approach that the authors use is quantitative convergent-beam electron diffraction (QCBED) and the system being studied here is FePd. While QCBED is a well-established diffraction technique, its applications for disordered crystals have been limited. Using QCBED, the authors have performed refinements of the disorder parameter, isotropic Debye-Waller factors (DWF) and the structure factors of (111) and (200) structure factors. The authors concluded that (1) disorder in FePd increases DWF, (2) has negligible influence on the bonding electron density, and (3) QCBED matches DFT and thus can be used to understanding bonding in chemically disordered systems.

While claim (1) is reasonable and expected, the claim (2) might be the result of limitations in the way the structure factors are probed, which needs further qualification. Claim 3 is encouraging and may further stimulate future studies on similar systems.

As written, the manuscript also contains several issues that need to be addressed.

First, the structure factors in equations 2 and 3 ignores the imaginary part from the electron absorptive potential. This part is sensitive to thermal vibrations and thus the temperature dependent part of DWF, while the static part from disorder should have very little effect. Thus, how absorption is dealt with is critical for the refinement.

Second, line 48, "disorder has proven very difficult as their interaction with the incident electron beam becomes increasingly weaker." This needs a clarification; it seems what the authors mean here is the decreasing intensity of order sensitive reflections.

Third, the authors have used relatively thick samples with thickness greater than 200 nm and reach ~400 nm. Even though the probe used is small (0.5 nm), the interaction volume is large and thus the measured structure factors is averaged over many unit cells, which is probably while the reported study is not sensitive to fluctuations in bonding charge density that may arise from disorder.

Fourth, the use of bonding electron density in the title implies a measurement of charge density, which is not the case here. The structure factors measured here are sensitive to bonding, but alone are not sufficient to describe the full 3D charge density. A modification is recommended here. Second, the disorder parameter in the title has a very specific meaning, e.g. in an intermetallic alloy, which should be made clear.

Fifth, it is not clear how to make sense of the comparison to DFT calculations. The supercells used for calculation no longer have the fcc lattice symmetry, is the fcc structure factor sufficient to describe the charges in a supercell? Does reflections along different directions differ? The results here also seem to not confirm any particular bonding configuration, which is in direct contradiction to the abstract claim that they revealed and confirmed the local bonding environment.

Other minor comments:

Line 48-51 This line is potentially misleading, since disorder also induces atomic displacements, which also impact structure factors and could dominate over the electronic structure part.

Line 79 – The way this is written makes it sound like it is the first time the DWF and structure factor have been refined. This is certainly not true, it might be the first time the disorder parameter has been refined.

Lines 164-182 These R_w values seem high and should be compared with other metrics such as chi-square and the standard R factor. Additionally, the R_w is calculated with the knowledge of intensity uncertainty (σ). How this was obtained should be specified in the methods. Related to this, how the intensity recorded after a GIF on CCD is processed should also be described, for example, the method used to correct the MTF or PSF of CCD.

Line 250 – There are several mentions of the probe size, but equally important is the convergence angle. This should be added to Table 1 to know how much of sample volume is covered.

Reviewer #3

(Remarks to the Author)

Version 1:

Reviewer comments:

Reviewer #1

(Remarks to the Author)

Dear Authors,

I am impressed with the rigour of the corrections that have been made and the explanations that have been directed to each of my points made in round 1 of refereeing. I thank the authors for their courtesy and care in responding.

The main issue that I had which centred on the assumption that F111 and F200 were enough to characterise the bonding electron density in FePd has largely been tackled and I am also happy that the authors have changed "bonding electron density" to "deformation electron density". This is more acceptable.

Some comments to follow up on your responses:

- I thought that superlattice reflections due to weak ordering might be present and you showed this nicely with the SAD pattern in Fig. R2. They are very weak indeed and so I can accept that they will make a minimal contribution. This has been quite nicely demonstrated.
- The approach of dealing with the PSF by convoluting the calculated patterns instead of deconvoluting the experimental patterns is very inventive and I like this approach. However, YOU USED A CCD ON A TRIDIEM GIF. This presents a significant oversight, i.e. that the PSF associated with CCDs has a very large and significant tail and that this is best modelled by a Lorentzian. In fact a standard multicomponent decomposition reveals that the Lorentzian is very significant and therefore the only sensible way of modelling a CCD PSF is the weighted combination of Gaussians and Lorentzians.

If I may be so bold as to suggest a simple solution for this issue: clearly, it is difficult to have a two-parameter optimisation of the Gaussian-Lorentzian function because of the nature of your piece-wise approach to the refinements. An easy way out of this is to measure your detector PSF accurately by the large range of simple methods available out there and then convolute the smoothed experimentally determined PSF with your calculations. This will negate you having to adjust two parameters in the PSF convolution step.

- To claim that distortions in the ZOLZ are linear is a first order approximation and it is good if you state this in the adjusted manuscript.
- Noise in CCD data is not just constant + Poisson. There is a huge addition due to gain noise since during each exposure, the detector itself changes state from the state it was in when it was gain referenced. Gain correction noise is linear and adds a significant fraction to the overall noise characteristics, especially when frame averaging (eg. Zuo, Ultramicroscopy 1996). You really need to say something about this or better still, do something about this.
- My main concern has been very well taken care of by Figs R22, R23 and R24. I will put this issue out of my mind as long as you include those figures and an associated explanation in the Supplementary Information.

IN SUMMARY:

I am happy to recommend acceptance, ON THE CONDITIONS:

- Please use a mixed Gaussian/Lorentzian OR experimentally measured PSF in all your pattern-matching analyses. The latter is better as it can account for detector anisotropy by not assuming that the PSF is radially symmetric.
- Please account for linear gain noise in your analyses.

- Please include Figs R22, R23 and R24 and a covering explanation somewhere either in the text or Supp Info, to assure the reader that the deformation electron density you are presenting is a close representation of the true bonding ED.

I am happy to look at these points again once the manuscript has been suitably revised.

Bravo - I look forward to seeing this article in print - these changes should be fairly easy to implement.

Sincerely, your reviewer.

Reviewer #2

(Remarks to the Author)

This revised version addressed the issues that were raised in my previous report. The authors' response is acceptable, except following which needs to be addressed:

Page 2: Lines 48-52. For the sentence of "remains challenging as their diminishing effects on measurable parameters such as electron density, electrostatic potential, and ordered-sensitive diffraction intensities", this is misleading, the disorder will increase diffuse scattering, but this is not measured here. The problem is associated with using Bragg diffraction alone to determine disorder effect, thus the problem is the sampling issue.

In "Crucially, all forms of disorder perturb electronic structures or even lattice displacements", "or even lattice" should be replaced by "and atomic", since it is atoms displace around defects, that can not be avoided.

Page 6: Lines 181-182/192-201. I'm not sure if I'm reading the contour plots in Fig.2j and Fig.2k incorrectly, but it looks like the minimized eta value is around 0.45 in the contour plots. These results either need an explanation or the previous values need to be reassessed considering relaxation of the DWF.

I think this is a major problem that needs to be addressed before publication. It is pretty clear the DWF is impacting the refinement. I think the plot needs to clearly state what the DWF used during refinement was. The contour plots clearly show local minima in the 0-0.5 range shown. Additionally, the R_w in these plots shows values at or below the determined minima for the refinement. It is possible that the graph is simply mis-labeled from 0-0.5 instead of 0-1.0, but this should be carefully checked.

Reviewer #3

(Remarks to the Author)

Version 2:

Reviewer comments:

Reviewer #1

(Remarks to the Author)

Dear Authors,

Thank you for treating all of my comments with great rigour and depth. The sensitivity tests that you have done with respect to different PSFs and incorporation of linear noise are very convincing and I believe that you have solid conclusions that are worth being published.

Bravo - I like this paper and look forward to seeing it in print.

The amendments more than satisfy me and I am actually quite surprised with the latest depth of treatment.

I recommend acceptance in this latest form without any further changes.

Yours sincerely, referee 1.

Reviewer #2

(Remarks to the Author)

I am satisfied with the revisions that authors have made in addressing the remaining issues pointed out in my last review.

Reviewer #3

(Remarks to the Author)

I co-reviewed this manuscript with one of the reviewers who provided the listed reports. This is part of the Nature

Communications initiative to facilitate training in peer review and to provide appropriate recognition for Early Career Researchers who co-review manuscripts.

Responses to Comments of NCOMMS-24-59352

We would like to thank the reviewer for the professional and insightful comments on our manuscript. The manuscript has been revised in accordance with the reviewers' suggestions, and we highlight the following improvements in the revised version.

- **CBED intensity data preprocessing:** We have described the process of CBED intensity data preprocessing, focusing on the correction of the background, the point spread function (PSF), and geometric distortions. These corrections led to a significant improvement of the average R_w from 0.3 to 0.2. As a result, the disorder parameter refined from both thin and thick regions is now more consistent, with patterns from thin areas showing greater sensitivity to the change of the disorder parameter.
- **Refinement algorithm:** We have elaborated on the refinement algorithm by discussing how uncertainties in CBED intensity were quantified, how the standard deviation of fitting results was calculated, and provided definitions for weighted R_w and normal R_w . The treatment of absorption factors is also discussed.
- **Correlation of fitting parameters:** We have explored the incremental impact of low-order structure factors on QCBED refinement, explaining why F_{200} was included in the refinement but later excluded in the construction of the deformation electron density map. The correlation between different fitting parameters is also discussed.
- **DFT calculation:** More details on how the deformation electron density maps were obtained for the supercells using DFT calculation are included.
- Grammatic mistakes have been corrected.

REVIEWER COMMENTS

Reviewer #1 (Remarks to the Author):

Comments on "Revealing disorder parameter and bonding electron density using electron diffraction" by Lin et al.

NCOMMS-24-59352

Summary:

The work presented introduces a disorder parameter, η , into the IAM structure factor calculation that is used to generate the reference structure factors in Bloch-wave calculations of CBED patterns being matched to experimental ones in a QCBED analysis of FePd. The bonding electron density is determined from the refined lower order structure factors and their differences from the IAM structure factors which incorporate the QCBED-optimised value of η .

The bonding electron densities obtained from QCBED-measured structure factors represent the average structure in the experimentally probed nanoscale volumes of disordered FePd and these are compared with DFT simulations.

This is a very interesting piece of work and a commendable direction for new research in QCBED, but there are many major scientific flaws that stem largely from an incomplete treatment of the problem and a lack of detail when it comes to treating systematic errors that are embedded in CBED data. The most fundamental error though, is one where the evidence that the authors have in front of them contradicts their approach to the problem. They have themselves concluded that Fe and Pd have different electronegativities and therefore, the approximation that FePd behaves as if it were an elemental metal in terms of bonding (allowing all structure factors other than F_{111} and F_{200} to be neglected in a bonding change density analysis), is fundamentally wrong. See points N and onwards below.

Reply: We appreciate the reviewer's positive comments on our work. We have addressed the reviewer's comments point by point in the following section.

The issues with this work and its presentation are numerous and significant and, in some cases, severe. I will outline these below in terms of issues with the manuscript and the problems with the science. The work requires major revision, but I do believe that if these revisions are carried out, it is certainly worthy of publication in Nature Communications and I support the authors wholeheartedly in their efforts to publish in this esteemed journal.

Reply: We appreciate the reviewer's insightful comments that helped us improve the manuscript.

The manuscript:

1. Page 1: I noticed that the author, Van Tendeloo is misspelled in the author list of the manuscript on page 1. This is very careless and is symptomatic of the many errors in the writing of the manuscript.

I advise the author (and the co-authors) to carefully proof-read manuscripts BEFORE submission. It is a pity that referees have to navigate so many mistakes which act as a fog in assessing work that does actually have potential.

Reply: We would like to thank the reviewer for pointing out all the careless mistakes in the manuscript. All the authors' names have been double-checked. We have proofread and corrected the grammar mistakes in the revised manuscript.

2. Page 2: "using the simultaneously imaging mode" on page 2 is poor English and incorrect grammar. There are many similar cases everywhere in the manuscript and I suggest that the authors engage the help of someone who writes English well to check their document. I am surprised that some of the co-authors did not pick these very obvious mistakes up in proof-reading.

Reply: In the revised manuscript, we have changed “using the simultaneously imaging mode” to “often integrated with transmission electron microscope (TEM) imaging modes”. The co-authors have also helped correct grammar mistakes. Please see page 2, line 58.

3. Page 2: *Two-beam is referred to incorrectly here. Very few true 2-beam studies have ever been published. The authors mean systematic-row QCBED, not 2- beam QCBED.*

Reply: We have changed ‘two-beam’ to ‘systematic-row’ in the revised manuscript. Please see page 2, line 61.

4. Page 2: ρ^{EXP} : *The "EXP" is not defined anywhere - do the authors mean experimental? If so, then equation 1 is not strictly an experimental total electron density because QCBED cannot measure all of the structure factors. It is possible to produce an experimental deformation electron density, $\Delta\rho^{EXP}(\mathbf{r})$, where QCBED measures all of the bonding-affected structure factors (and thus all the IAM structure factors cancel out in producing the difference, but it is not possible for QCBED to determine a total bonding electron density as it would have to measure all structure factors to high $\sin\theta/\lambda$.*

Reply: We would like to thank the reviewer for the valuable suggestions. The reviewer is correct that only low order structure factors, or bonding-affected structure factors can be measured using QCBED. We have replaced $\rho^{EXP}(\mathbf{r})$ with $\Delta\rho^{EXP}(\mathbf{r})$, and defined EXP in $\Delta\rho^{EXP}(\mathbf{r})$ as “experimental deformation electron density” in the revised manuscript. Please see page 3, line 64.

5. End of page 2: *"via Fourier transform" is poor English.*

Reply: In the revised manuscript, we have rephrased this sentence and avoided using “via Fourier transform”. Please see page 3, line 66.

6. Page 3, line 65: *"has" should be "have" as this is currently a singular/plural*

contradiction.

Reply: We have changed ‘has been’ to ‘have been’ in the revised manuscript. Please see page 3, line 70.

7. Top of page 3: The first sentence is wrong. As said before, QCBED cannot measure $\rho^{EXP}(r)$. It has been used to determine $\Delta\rho^{EXP}(r)$ and the cited publications bear this out.

Reply: We have replaced $\rho^{EXP}(r)$ with $\Delta\rho^{EXP}(r)$ in the revised manuscript. Please see page 3, line 69.

8. Page 4, line 102: "MBOZA condition means..." is a bad way to start a sentence. This mode of expression is poor English.

Reply: We have replaced “MBOZA condition means...” with “Under MBOZA conditions” in the revised manuscript. Please see page 4, line 102.

9. Page 4, lines 107-108: "whether the MBOZA condition is also sensitivity to change..." - again poor English. You mean "sensitive".

Reply: We have corrected this mistake in the revised manuscript. Please see page 4, line 107.

10. Page 4: I am certain that η is missing somewhere in equation 2. It is in equation 3 and looking at the dependence of F_g ($h + k + l$ even) on η in Fig. 1c, it is obvious that η has been left out of equation 2. This is extremely careless and I am astonished that this was missed in proof-reading the manuscript.

Reply: We agree with the reviewer that it is a bit surprising that F_g^X ($h+k+l$ even) does not depend on η . The detailed calculation is as follows.

There are four atoms in the tp2 cell: Pd atom at (0 0 0) with occupancy $\frac{\eta}{2}$; Pd atom at (0.5 0.5 0.5) with occupancy $1 - \frac{\eta}{2}$; Fe atom at (0 0 0) with occupancy $1 - \frac{\eta}{2}$; Fe atom at (0.5 0.5 0.5) with occupancy $\frac{\eta}{2}$;

$$\begin{aligned}
F_g^X &= \sum_i f_i(s) e^{-B_i s^2} e^{-2\pi i g \cdot r_i} \\
&= \left[\left(1 - \frac{\eta}{2}\right) f_{Fe}(s) e^{-B_{Fe} s^2} + \frac{\eta}{2} f_{Pd}(s) e^{-B_{Pd} s^2} \right] e^{-2\pi i (h \cdot 0 + k \cdot 0 + l \cdot 0)} + \left[\frac{\eta}{2} f_{Fe}(s) e^{-B_{Fe} s^2} \right. \\
&\quad \left. + \left(1 - \frac{\eta}{2}\right) f_{Pd}(s) e^{-B_{Pd} s^2} \right] e^{-2\pi i (h \cdot 0.5 + k \cdot 0.5 + l \cdot 0.5)} \\
&= \left(1 - \frac{\eta}{2}\right) f_{Fe}(s) e^{-B_{Fe} s^2} + \frac{\eta}{2} f_{Pd}(s) e^{-B_{Pd} s^2} + \left[\frac{\eta}{2} f_{Fe}(s) e^{-B_{Fe} s^2} \right. \\
&\quad \left. + \left(1 - \frac{\eta}{2}\right) f_{Pd}(s) e^{-B_{Pd} s^2} \right] (-1)^{h+k+l}
\end{aligned}$$

when $h + k + l$ is odd:

$$\begin{aligned}
F_g^X &= \left(1 - \frac{\eta}{2}\right) f_{Fe}(s) e^{-B_{Fe} s^2} + \frac{\eta}{2} f_{Pd}(s) e^{-B_{Pd} s^2} - \frac{\eta}{2} f_{Fe}(s) e^{-B_{Fe} s^2} \\
&\quad - \left(1 - \frac{\eta}{2}\right) f_{Pd}(s) e^{-B_{Pd} s^2} = (1 - \eta) (f_{Fe}(s) e^{-B_{Fe} s^2} - f_{Pd}(s) e^{-B_{Pd} s^2})
\end{aligned}$$

when $h + k + l$ is even:

$$\begin{aligned}
F_g^X &= \left(1 - \frac{\eta}{2}\right) f_{Fe}(s) e^{-B_{Fe} s^2} + \frac{\eta}{2} f_{Pd}(s) e^{-B_{Pd} s^2} + \frac{\eta}{2} f_{Fe}(s) e^{-B_{Fe} s^2} \\
&\quad + \left(1 - \frac{\eta}{2}\right) f_{Pd}(s) e^{-B_{Pd} s^2} = f_{Fe}(s) e^{-B_{Fe} s^2} + f_{Pd}(s) e^{-B_{Pd} s^2}
\end{aligned}$$

The results are the same as Equations (2) and (3) in the manuscript. For clarity, we have now included the detailed calculation in the Supplementary information as Supplementary Note 2.

11. Page 4: Supplementary Fig. 1 is totally superfluous. All one ever needs to say is where the scattering factors came from. You audience does not need the obvious shown to them - they all know what scattering factors look like as a function of s .

Furthermore, when citing from where the scattering factors came from, reference 36 is far too general and not specific enough. You need to specify that they are in volume C of the International Tables for Crystallography. Furthermore, at this point, you should actually cite Doyle and Turner (ref. 38) first as they originated these scattering factors. Reference 36 should be mentioned as a secondary reference.

Reply: We have removed Supplementary Figure 1 as suggested by the reviewer, and cited the references as recommended by the reviewer, in the revised manuscript. Please see page 4, line 115.

12. References section: There are few duplicated references. Please check your references section for duplicates.

Reply: We would like to thank the reviewer for the careful observation. We have combined 2 duplicated references (*Science*, 2011, 331(6024): 1583-1586) in the revised manuscript.

13. Page 4, lines 114-116: "...plotting the difference ΔF_g^X ..." is a tautology because ΔF_g^X is already a difference. If you want to say the word difference, you would say "...plotting the difference in structure factors, ΔF_g^X ,..."

You are claiming that the difference being plotted is between $F_g^X(B_{iso}, \eta, s)$ and $F_g^X(0, 1, s)$ but this is an incomplete description because the plot in Fig. 1c shows only cases where $B_{iso} = 0$. Please make your description more accurate.

Reply: We agree with the reviewer that only two special cases are considered in Fig. 1c and Fig. 1d. In the revised manuscript, the description has been changed to "The influence of η on $F_g^X(B_{iso}, \eta)$ is visualized by plotting the difference between $F_g^X(0, \eta)$ and $F_g^X(0, 1)$ with B_{iso} fixed at 0 (Fig. 1c). Here $F_g^X(0, 1)$ is the structure

factor of the completely disordered ($\eta = 1$) and static ($B_{iso} = 0$) tp2 cell” and “Similarly, the influence of B_{iso} is visualized by plotting the difference between $F_g^X(B_{iso}, 1)$ and $F_g^X(0, 1)$ with η fixed at 1 (Fig. 1d)”. Please see page 4 lines 116-124.

14. Fig. 1 caption: s is not the scattering vector!! It is $|g|/2$. s is a scalar, NOT a vector. This manuscript is very poorly written.

Reply: We have removed the “scattering vector” when describing s in the revised manuscript. s is defined as $\sin\theta/\lambda$ or $|g|/2$ and then referred as “ s ”. Please see page 4, line 113.

15. Fig.1 caption: The last part of the last sentence should be rewritten as it contains a singular/plural contradiction.

Reply: The last sentence in Fig.1 caption is revised to “Grey rectangular shadows in (c) and (d) highlight the behaviour of low-order F_g^X from (001) to (222) with s ranging from 0.131 to 0.455 \AA^{-1} ”. The entire figure caption has been rephrased for better clarity. See page 5, lines 129-135.

16. Page 5, line 136: Missing "the" before "Ewald sphere".

Reply: We have added “the” before “Ewald sphere” in the revised manuscript. Please see page 5, line 137.

FROM HERE ONWARDS, I WILL NO LONGER HELP WITH CORRECTING BASIC ENGLISH AND GRAMMAR - PLEASE ENGAGE SOMEONE TO HELP TIDY UP YOUR ENTIRE MANUSCRIPT.

17. Page 5, lines 140-143: The sentence "The (0kl)-type disks with odd $k + l$...or 'disordered' CBED pattern (Fig. 2d)." requires an explanation. It is simply not appropriate to make the observation of absent disks without trying to furnish the

observation with any kind of explanation.

Reply: We agree with the reviewer that more analysis is needed to support this claim. Figure R1 shows that the average intensity from the (001) disc is 66, while the background intensity is 70, proving that the (001) disc has an average intensity comparable to the background. The slight ordering, however, can be observed in the selected area electron diffraction pattern (see the reply to comment ‘A’ and Figure R2). We have revised the sentence in the manuscript to “Selected area electron diffraction (SAED) acquired from the disordered sample indicates that the intensity of ‘ordered’ reflections ($h + k + l$ odd) is around 0.1% of the main reflections (Supplementary Fig. 1). In the two ‘disordered’ CBED patterns, the ‘ordered’ discs have intensity comparable to the background (Supplementary Fig. 2)” and included Figures R1 and R2 as Supplementary Figure 1-2. Please see page 5, lines 142-146.

Figure R1 Comparison between the average intensity from the (001) disc (white dashed rectangle) and the background (yellow dashed rectangle) of the CBED pattern shown in Fig. 2c.

18. Throughout the manuscript, the authors label structure factors as F_{ghkl} . This is incorrect. Given that g is defined by hkl , then it suffices to write F_{hkl} and it is incorrect to write it otherwise as to do so would be a symbolic tautology.

Reply: We agree with the reviewer that F_{hkl} is sufficient to describe the structure

factor F_g , since g is defined by hkl . We have replaced F_{ghkl} with F_{hkl} in the revised manuscript.

19. *Caption to Fig. 3: "a completed disordered MBOZA CBED pattern." This implies that the CBED pattern is disordered!! Furthermore, the grammar is incorrect - I think you mean "completely disordered". It should be rewritten as "an MBOZA CBED pattern from a completely disordered region of the specimen."*

Reply: We have revised the figure caption as suggested by the reviewer. Please see page 10, line 243.

20. *Page 10, line 248: What do you mean by saying that the CBED patterns were statistically analysed? Don't you mean that the results from QCBED of the 8 patterns were statistically analysed?*

Reply: The reviewer is correct that the results from the 8 CBED patterns were statistically analysed, not the CBED patterns. Section 2.2 has been completely rewritten in the revised manuscript to answer the reviewer's comments on the refinement process. We have revised the sentence to "Eight MBOZA CBED patterns acquired at -170 °C were systematically refined using the incremental approach (Table 1, Fig. 3e-h, Supplementary Table 1-4)". Please see page 10, line 252.

21. *Page 10, line 254: You should quote the results in the reverse order because the sentence leads one to believe that you are assigning the first result in parentheses to the larger probe size (mentioned most recently). This way of quoting results is confusing and reflects poor writing.*

Reply: We would like to thank the reviewer for noticing this error. The sentence has been revised to " B_{iso} from the thicker region (0.475(9)) exhibits smaller uncertainty than the thinner region (0.48(4))" in the revised manuscript. Please see page 11, lines 262-263.

22. Page 10, line 255: "more different" is a poor conjunction in English. Please re-write this part.

Reply: We have replaced "more different" with "likely due to the larger probe size and larger thickness capturing a greater variety of local configurations and enhancing statistical consistency" in the revised manuscript. Please see page 11, line 264.

23. Page 11, line 278: Within 0.1% error of what reference? If one claims an error, one must state unequivocally what the error is relative to.

Reply: We agree with the reviewer that more details need to be discussed on this "0.1% error". Here the error means the difference between lattice parameters a , b , and c . If the $4 \times 4 \times 4$ supercell is not random enough, a , b , and c would be noticeably different. Using the SQS method, the lattice constants ($a = 15.235 \text{ \AA}$, $b = 15.249 \text{ \AA}$, and $c = 15.238 \text{ \AA}$) are within 0.1% error compared with each other, indicating that the distribution of Fe and Pd atoms is sufficiently random. We have clarified this point in the revised manuscript. Please see page 12, lines 284-286.

24. Only in the caption to Fig. 4 is DFT mentioned for the first time. This should be fully explained in the text as per point L below.

Reply: In the revised manuscript, DFT is mentioned and defined in the introduction (page 3, line 71). We also discussed the DFT optimization details in "Methods". Please see page 12, line 283.

25. Fig. 4, page 12 & line 320 on page 13: Why are only some of the tetrahedral interstices and only some of the octahedral interstices shaded in figure 4f?

Reply: We agree with the reviewer that this may cause confusion. It would be not easy to distinguish different tetrahedrons or octahedrons if they were all shaded. In Fig. 4f of the revised manuscript, we have added grey and yellow spheres to indicate the locations of tetrahedral interstices and octahedral interstices, respectively.

26. Page 13, line 316: *The difference structure factors are missing units.*

Reply: We would like to thank the reviewer for pointing this out. We have now added the unit “e⁻/atom” for structure factors throughout the manuscript, similar to the unit used in the literature (*Science*, 2011, 331(6024): 1583-1586).

Scientific issues:

A. *"For $\eta = 0$, the structure is fully chemically ordered, while $\eta = 1$ indicates complete chemical disorder." What if $Occ(Fe_{Pd}) = 0.5$ in a manner where a perfectly ordered superlattice is formed? In that case, the "disorder" parameter $\eta = 1$ but you could still have a perfectly ordered and periodic superlattice. This means that your definition of the disorder parameter is not canonical.*

Furthermore, you would see the presence of a superlattice in your diffraction patterns. Do you see any extra reflections in some of your CBED patterns?

Reply: This is an interesting question. The review suggested the possibility that Fe and Pd atoms are not randomly distributed at (0 0 0) or (0.5 0.5 0.5). In that case, an ordered, periodic superlattice with a unit cell larger than the tp2 cell is formed. As the reviewer mentioned, such superstructures have been commonly observed in other intermetallics and can cause extra diffraction spots. Figure R2 shows a typical diffraction pattern acquired along [100] zone axis for a tp2 cell (or [110] zone axis of disordered fcc FePd). The left part is the diffraction pattern with normal contrast, and the right has enhanced contrast to show weak diffraction spots due to the slight tp2 ordering. The line profile shows that main reflections such as (011) spots have intensities around 4000, while the (001) spot has an intensity of around 4, which is only 0.1% of the main reflections. All the weak diffraction spots are from tp2 ordering, indicating that any other long-range ordering should be negligibly weak. We have included Figure R2 as Supplementary Figure 1 to clarify this comment from the reviewer, and also as an example to show that the disordered FePd is actually slightly ordered, which motivates us to measure the

disorder parameter using QCBED.

Figure R2 A typical diffraction pattern acquired along $[110]$ zone axis of disordered fcc FePd (or $[100]$ zone axis for a tp2 cell).

B. Figure 1: The drawing in part b is incorrect. The transmitted beam (or 000) is always in the ZOLZ, never the FOLZ. The Laue circle of Bragg-satisfied reflections that includes the central beam, 000, is always part of the ZOLZ by definition.

THE ERRONEOUS USE OF FOLZ INSTEAD OF ZOLZ OCCURS IN MANY OTHER PLACES IN THE MANUSCRIPT. PLEASE FIX THIS.

Reply: We thank the reviewer for pointing out this error. In the revised manuscript, we have changed “FOLZ” to “ZOLZ” for every occurrence in the text and the figure.

C. Figure 2: In the pattern from the disordered material, there appears to be no noticeable disk perimeter. This may be symptomatic of the CBED probe averaging over a range of varying lattice parameters. What has been done within the QCBED refinement code, MBFIT, to accommodate variations in, or averaging over, fluctuating lattice parameters?

This constitutes a major systematic error that must somehow be accounted for within the refinement algorithm and/or processes.

Reply: We agree with the reviewer that the disordered structure should have a local variation of lattice parameters, which could influence refinement results. The disc perimeter is not as blurred as the reviewer might think. Figure R3 shows the CBED pattern used in Figure 2d but with enhanced contrast. The edges are clear for regions that are indicated by red arrows. For the regions indicated by green arrows, the edges blend into the background because the diffraction intensity over there is very low. The disc in the centre is very dark because it deviates greatly from the Ewald sphere.

Figure R3 A CBED pattern with enhanced contrast showing the discs' edges.

We agree with the reviewer that lattice constant variation can cause a blurring effect, which can also be caused by other factors such as thickness variation and point spread function (PSF). We do not distinguish between these different blurring factors during QCBED refinement. Instead, the overall blurring effect is considered by convoluting the calculated CBED pattern using 2D Gaussian distribution with different

σ . We then test if R_w can be improved as σ varies, and if the convolution influences the refinement results. Because both reviewers have further comments on the PSF, it is better to explain all the details here.

One more parameter needs to be mentioned, namely, the grid size N . From the excited-row method to MBOZA method, the refined data points change from 1D to 2D, and a big challenge is the computation time. For example, each MBOZA CBED disc in Figure R3 contains 147847 pixels. It is very time-consuming to calculate the diffraction intensity of every pixel by solving the Schrodinger equation in the periodic potential with a different beam-sample orientation. A conventional approach is to simulate data points for a $N \times N$ grid, and then interpolate the data points in between the grid points. The number N can be determined by performing QCBED refinement using different N and then checking which number yields the lowest R_w value. Figure R4a shows that as N changes from 31 to 101, R_w first decreases, reaches a minimum for $N = 61$, and then slowly increases again. For $N = 31$, the grid size is too small, and the calculation is significantly under-sampling (see Figure R4b), resulting in a very large R_w . As N further increases beyond 61, the computation time scales with N squared, but R_w did not get any better. Therefore, when we first started the project, the computing power was limited, and we chose $N = 61$ as a compromise between time and accuracy for previous refinement.

Figure R4 (a) R_w as a function of grid size N for the CBED pattern in Fig. 2d. (b) Experimental CBED discs (first row), calculated CBED discs (second row), and the absolute different (third row), of the QCBED refinement results using grid size $N = 31$.

For $N = 61$, as σ increases, convoluting calculated CBED pattern first reduces R_w and then worsens R_w (Figure R5). This is because the sampling rate is not sufficiently high. The convolution is influenced by pixels that are too far away, which can deteriorate refinement results for large σ .

Figure R5 R_w as a function of σ for grid size $N = 61$.

A similar trend has been observed for $N = 101$. As σ increases, R_w rapidly decreases and then slightly increases to a stable value (Figure R6). For $N = 201$, R_w does not increase as σ further increases, probably due to the high sampling rate. The

simulated and experimental line profiles are also compared for some combinations of N and σ (Figure R7). For $N = 31$, some small peaks are significantly blurred due to undersampling (Figure R7a). For $N = 101$ and $\sigma = 0$, the calculated peaks have higher intensities than the observed peaks, due to the various blurring effect (Figure R7b). For $N = 101$ and $\sigma = 1$, the intensities of the calculated peaks are lowered, resulting in between match between observed and calculated peaks. Therefore, the introduction of σ can indeed broaden the simulated CBED patterns to improve the fitting results. In conclusion, as $N = 201$ does not noticeably improve R_w while consuming 4 times more computing power, we decided to use $N = 101$ and $\sigma = 1$ for QCBED refinement in the revised manuscript.

Figure R6 R_w as a function of σ for grid size (a) $N = 101$ and (b) $N = 201$.

Figure R7 Line profile comparison for (222) disc calculated using different N and σ .

Figure R8 (a) B_w , (b) F_{111} , and (c) F_{200} for different N and σ . The dashed lines indicate IAM values.

Figure R8 summarizes the fitting results of B_{iso} , F_{111} , and F_{200} for different N and σ . B_{iso} varies from 0.457 \AA^2 to 0.472 \AA^2 , F_{111} varies from $27.159 \text{ e}^-/\text{atom}$ to $27.166 \text{ e}^-/\text{atom}$, and F_{200} changes from $25.542 \text{ e}^-/\text{atom}$ to $25.545 \text{ e}^-/\text{atom}$, clearly indicating that the fitting results are insensitive to the blurring from various factors. Therefore, the blurring effect is not a main source of error in QCBED refinements. This brings up another advantage of using the MBOZA condition, that the fitting results seem to depend on the locations of the features, instead of the intensity of the features. The discs under MBOZA condition have much more features than other beam-sample orientations, which is important for robust refinement results. Figure R9 shows that the zone axis pattern has a very bright 000 disc while other diffraction discs are all very weak, as confirmed by the line profile. The features in all the discs are relatively simple with few peaks, indicating that the dynamic diffraction is not strong enough. On the contrary, under the MBOZA condition, the diffraction discs are much brighter, and the

features are more complex. Due to strong dynamic diffraction, more electrons are scattered to diffraction discs. The larger number of peaks within the features makes this pattern more robust against the influence of background, PSF, and distortion.

Figure R9 Comparison between a zone axis pattern and an MBOZA condition pattern acquired from L1₀ FePd [101] zone axis.

To summarize, the overall blurring effect is considered by convoluting the simulated CBED pattern instead of deconvoluting the experimental CBED pattern. For fast CBED refinement, using a small grid size ($N = 61$) is sufficient, and the refinement results are very similar to those using a large grid size. Small grid size is therefore useful to analyse the correlation between different refinement parameters when many refinements are needed (see the reply to comment 'F' for example). For the data points reported in the revised manuscript, $N = 101$ and $\sigma = 1$ are used to minimize systematic error caused by the overall blurring effect. We have included the discussion on PSF on page 6 lines 155-158, and Supplementary Figures 7-9.

D. After line 146 on page 5, it becomes clear that η is not a freely refineable parameter within the QCBED software that has been applied, i.e. MBFIT. This means that the

authors have changed η and re-refined for fixed values of this parameter. While this affords the reduction of parameter correlations with respect to this parameter, it does not allow it to be fine-tuned in a continuous fashion. This is a weakness that needs to be overcome while at the same time, developing a refinement scheme that subdues parameter correlations.

Reply: The reviewer is correct that η is not a freely refinable parameter within the MBFIT software. In the revised manuscript, the R_w value was calculated using much denser η values with small step size. The optimized η value can be obtained by fitting the $R_w - \eta$ curve near the minimum using quadratic polynomials. For the two CBED patterns used in Fig. 2c-d, we can obtain the η to be 0.855 and 0.856, respectively (Figure R10). The fitting results are also included in Supplementary Figure 10. We have optimized η using this approach in the revised manuscript. Please see page 7, lines 180-182.

Figure R10 The fitting of η using $R_w - \eta$ curve.

We then test the validity of this approach using a parameter that can be directly refined using the MBFIT software. In the example, the R_w was obtained by fixing F_{110} while relaxing all the other parameters in the refinement. The $R_w - F_{110}$ curve is shown in Figure R11. Quadratic polynomial fitting suggests that the refined value is 54.302 for F_{110} , which agrees very well the value 54.317 when F_{110} is relaxed along all the other parameters. This example suggests that this approach can be used to determine parameters that are difficult to be implemented in the refinement software.

Figure R11 The fitting of F_{110} using $R_w - F_{110}$ curve.

E. Very few CBED patterns are actually refined in this work which does not allow for an assessment of the reproducibility of the results.

This work would be more significant if the authors were to produce a map of disorder by measuring η from say 400 CBED patterns collected in a grid of 20×20 points spanning the spatial range of order fluctuations within the specimen.

Reply: We agree with the reviewer that acquiring CBED patterns from a grid is an exciting idea. To explore this possibility, we have written a DigitalMicrograph script to acquire a series of CBED patterns while the beam is scanning. The acquisition process was captured and included as Supplementary Video 1 for review only. Supplementary video 2 for review only shows the 25 CBED patterns collected in a grid of 5×5 points. The exposure time for each pattern is 4 s. The 25 CBED patterns are put together in one figure, which is shown below as Figure R12. We have refined all 25 patterns and obtained the thickness mapping and the structure factor mappings (Figure R13). The η mapping is obtained by simply finding the minimum of the $R_w - \eta$ curve for each CBED pattern. Therefore, in principle, the mapping of disorder parameter could be obtained using this method.

Figure R12 25 CBED patterns sequentially collected in a grid of 5x5 points.

Figure R13 Mapping of the thickness, the disorder parameter, and structure factors.

However, there are several reasons that this approach is not perfect yet. First, the FePd sample was prepared using jet polishing to obtain a smooth surface; this, however leaves the sample significantly bent. For example, the 1st CBED pattern and the 25th CBED pattern from the 5x5 grid show different orientations, as indicated by the Kikuchi lines and the intersection between the Ewald sphere and the ZOLZ (Figure R14). Here, the Kikuchi lines are superimposed on the CBED pattern using a program called AbjCbd that we developed to preprocess CBED patterns. The locations of Kikuchi lines are determined by examining the symmetry of each disc. For example, the (022) disc has a 2-fold axis, while the (020) and (002) discs have mirror symmetries. When the beam-sample orientation deviates from the MBOZA condition, the fitting result deteriorates very fast. Therefore, it is very difficult to acquire a large number of CBED patterns from a large field of view, at least for the disordered FePd sample.

Figure R14 Orientation difference between the (a) first frame and (b) the last frame.

Second, the sample has many planar defects and dislocations (**Figure R15**). Whenever the CBED pattern is acquired near the defects, the local strain can cause symmetry breaking of some discs that are precisely on the Ewald sphere (**Figure R15**). These CBED patterns tend to have high R_w . This limits the scanning area to the order of tens of nm, which does not give us much information on the spatial distribution of the disorder parameter. This 4D CBED approach inspired by the reviewer will undoubtedly be used to investigate other appropriate sample systems, but much work still needs to be done before this 4D CBED data can be published with sufficient confidence. Therefore, we decide not to include this data in the current manuscript.

Figure R15 (a) Dislocations in the FePd sample and (b) symmetry breaking in a CBED pattern.

F. The R_w results do not change much with increasing η above about 0.6 as the authors correctly state. In QCBED, geometric distortions in the data need to be fully accounted for because these have a much larger effect on R_w than the small changes due to η .

In the absence of any discussion at all on the effects of geometric distortions on the QCBED results, it is highly doubtful that there is much weight in the statement "CBED refinement is generally sensitive to η ".

Reply: We agree with the reviewer that geometric distortion from the GIF system needs to be discussed in the manuscript. Because only low order discs are included in the refinement, we assume that geometric distortion is linear, which is quite different from the case of HOLZ discs where nonlinear distortions need to be considered (*Acta Crystallographica Section A: Foundations of Crystallography*, 1999, 55(5): 939-954). Figure R16 shows a typical CBED pattern being preprocessed using the software AdjCbd to generate intensity data for MBFIT refinement. The locations of the centre disc, the two nearest discs R_1 and R_2 , and the Kikuchi pattern can be freely adjusted. For this example, the mirror symmetries in (0-20) and (002) discs, and the 2-fold symmetry in the (0-22) disc can be used to guide the location of discs and Kikuchi lines.

We then determine the angle and ratio between R_2 and R_1 , and compare them with standard lattice constant to evaluate the linear distortion. For this particular CBED pattern (Figure R17), R_1 is (-208.3 160.9) while R_2 is (232.3 305.8), which yields $R_2/R_1 = 1.459$ and the angle is 89.5° . Using the standard lattice constants, $a = 2.692 \text{ \AA}$ and $c = 3.807 \text{ \AA}$, the ratio should be 1.414 and the angle should be 90° . We then consider this geometric distortion when outputting the intensity data from this CBED pattern. We have added relevant discussion in the revised manuscript on how the geometric distortion is considered. Please see page 6, line 155.

Figure R16 The AdjCbd software interface showing the location of discs and Kikuchi lines. The two nearest vectors, R_1 and R_2 , are indicated by yellow arrows.

Figure R17 The geometric parameters panel for the CBED pattern shown above.

Further on this same point, the Methods section gives no information as to how any of the systematic and random errors in the CBED data were dealt with. For example:

• *How was the point spread function measured and removed?*

Reply: As mentioned in the reply to the reviewer's comment 'C', instead of trying to deconvolute experimental CBED patterns, the overall blurring effect from the PSF and other factors is considered by convoluting the calculated CBED pattern with a 2D Gaussian distribution of different σ . The details are discussed when replying to comment 'C'.

• *How was the noise as a function of signal determined? Note that most detectors are never Poisson so to assume that $\sigma_i \propto I_i^{obs}$ in equation 6 in the Methods would be very wrong. This alone can result in refined parameters in shallow parts of the R_w surface in the N -dimensional parameter space of a QCBED refinement, being in greater error than the differences the authors claim to be resolvable when it comes to determining the optimal η from the more disordered regions.*

Reply: We agree with the reviewer that this part needs to be clarified. The uncertainty or the noise depends mainly on the three parts. The first part is the photon noise σ_p (also called shot noise) that follows the Poisson distribution. σ_p is therefore proportional to $\sqrt{I_i^{obs}}$. The second part is dark noise that depends mainly on the device temperature. This part can be significantly reduced by cooling the CCD, a routine for Gatan CCDs equipped on most microscopes. The third part is the read noise that originates from the analog-to-digital conversion. This noise is added uniformly to every pixel. Therefore, the noise combines the shot noise and a uniform background noise σ_{bkgd} . In summary, for each pixel, the overall noise σ_i is defined as $\sigma_i^2 = \sigma_{ip}^2 + \sigma_{bkgd}^2$ where $\sigma_{ip} = \sqrt{I_i^{obs}}$. This approach has also been used in CBED refinement by other authors (*Acta Crystallographica Section A: Foundations of Crystallography*, 1995, 51(1): 7-19). The background is determined by measuring the average intensity of the region between two adjacent discs (Figure R18).

In the previous version of the manuscript, we ignored the influence of the constant

background on σ_i , which is the main reason we got a high R_W value. We have corrected this in the revised manuscript. Combined with the influence from PSF, the general R_W value is now reduced from 0.3 to 0.2. The correct treatment of σ_i is now included in “Methods”. Please see page 18, line 446.

Figure R18 Background determination from a CBED pattern. The right side is the line profile from the area indicated by the red arrow.

Continuing on point F, the authors later on page 6 state that the "CPA does not fully capture...". While these statements are true, the authors may not be getting the most out of their application of the CPA because they have compromised the potential of what they have set out to do by not dealing with all aspects of the data collection, processing and refinement as accurately as possible.

In the above light, to claim "the disorder parameter of the three regions was successfully measured using CBED...", at the bottom of page 6, is highly dubious.

Reply: We agree with the reviewer that it might be unfair to suggest that CPA is the main reason that R_W deteriorates. By properly removing the effect of background and the PSF, and assigning the right values for the noise, R_W has decreased from 0.3 to 0.2 for the eight CBED patterns reported in the manuscript. As a result, we are now more confident that the disorder parameter can indeed be refined from CBED patterns.

In addition, in the revised manuscript, we also included an analysis of the coupling between the disorder parameter, DWF, structure factors, and their influence on R_W , to

support the conclusion that “the disorder parameter was successfully measured using CBED”. To claim that these parameters can be simultaneously determined, it is important to prove that they are not correlated and that, there is a well-defined global minimum in a multi-variant space. Take η and B_{iso} for example; we performed QCBED refinement using fixed η from 0 to 1 with a step size 0.1, and fixed B_{iso} from 0.1 to 1.1 Å with a step size of 0.1 Å. The resulting R_w as a function of η and B_{iso} can be used to construct an isosurface plot (Figure R19), which has a well-defined global minimum. The R_w isosurface of $\eta - B_{iso}$, $\eta - F_{110}$, $\eta - F_{002}$ was calculated for the three CBED patterns in Fig. 2b-d and summarized in Figure R19. In all nine cases, the isosurface is relatively isotropic around the global minimum, indicating that these parameters are not strongly coupled with each other. The discussion on the isosurface plots is now included in the revised manuscript, and the isosurface plots of $\eta - B_{iso}$ are now included in Fig. 2i-k. Please see page 7, lines 192-201.

Figure R19 The R_w isosurface of (a-c) $\eta - B_{iso}$, (d-f) $\eta - F_{110}$, (g-i) $\eta - F_{002}$ of the three CBED patterns in Fig. 2b-d.

G. If F_{220} was refined as stated on page 8, line 213, then why are only graphs of refined F_{111} and F_{200} versus thickness shown in Fig. 3 (e & f)? For completeness, the graph of refined F_{220} versus thickness should also be shown.

Reply: We agree with the reviewer that details regarding F_{220} need to be clarified. We notice that the reviewer has further comments regarding the inclusion of F_{220} in the bonding electron density calculation. We will include more details when replying to those comments. The F_{220} plot is now included in Fig. 3h as suggested by the reviewer.

H. At the bottom of page 9 and over onto page 10, the following statement is made: "A better agreement, achieved by relaxing the low order F_g in the first refinement, verifies that the low order F_g contribution encoded in the CBED pattern can be extracted using MBOZA CBED technique and the CPA approach for a disordered structure."

This statement is wrong because all that has been shown is that there is an improvement in fit compared to the fit between the IAM pattern and the experimental one. This will always be true regardless of the veracity of the parameters that have been adjusted. You will always get a better fit than for the IAM, for any parameter you switch on in addition to thickness because the freely adjustable parameter will simply act as a buffer.

You need to develop a very solid refinement strategy that reduces parameter correlation and returns meaningfully interpretable parameters. The statement quoted above is a statement of the obvious and does not lead to anything beyond the obvious. It also does not allow much meaning to be drawn from the refined values of the parameters because the manner of the refinement is absent from such a statement.

Reply: We agree with the reviewer that more details need to be provided on the refinement strategy to avoid this type of confusion. Although it seems that the inclusion

of one more parameter can always improve the fitting results, there is also a limit beyond which the fitting cannot be improved anymore. Our emphasis on the effective decrease of R_w is based on two observations from our previous experience with CBED refinement. First, for CBED patterns under some beam-sample orientations, switching on low order structure factors in the refinement cannot noticeably reduce R_w . Second, as more structure factors are included in the refinement, R_w seems to reach a plateau and cannot decrease anymore. Take B_{iso} and low order structure factors of the disordered fcc structure as an example (Figure R20). The first refinement was performed by only relaxing B_{iso} . Then the first low order structure factor F_{111} is included in the refinement, which reduces R_w by around 0.01. The inclusion of F_{200} further reduces R_w by around 0.06. After that, the inclusion of F_{220} and F_{113} in the refinement leads to a negligible decrease of R_w . Therefore, the refinement is only sensitive to the first several low order structure factors. This trend has also been reported for other ordered materials (*Acta Crystallographica Section A: Foundations of Crystallography*, 2011, 67(3): 229-239). In the revised manuscript, instead of mentioning the IAM case and F_g -relaxed case, we now discuss this incremental refinement approach in detail. Please see page 9, lines 226-241. One subplot from Figure R20 is included in Fig. 3c, and the rest is included in Supplementary Figure 12 for clarification.

Figure R20 The change of R_w as more structure factors are relaxed in the QCBED refinement for the 8 CBED patterns used in the manuscript.

I. Does beam damage play a role in the collection and analysis of the data? The authors should give the threshold energies for ballistic damage for both iron, palladium and the ordered intermetallic FePd. The latter can be estimated if no specific measurements exist in the literature.

Reply: Beam damage does not play a significant role here. For example, we acquired 20×20 CBED patterns with an exposure time of 4 s from a $500 \text{ nm} \times 500 \text{ nm}$ region at liquid nitrogen temperature. STEM images acquired before and after the acquisition

do not show noticeable sign of beam damage (Figure R21). Normally the beam does not dwell at the same region for that long period of time.

Figure R21 Comparison between STEM images captured before and after acquiring 400 CBED patterns.

The maximum elastic collision energies (E_{max}) transferred from the electron beam to an atom can be calculated using the following equation (*Micron*, 2004, 35(6): 399-409; *Nature Materials*, 2021, 20(7): 951-955):

$$E_{max} = \frac{2ME(E + 2mc^2)}{(M + m)^2c^2 + 2ME}$$

where M and m refer to the mass of the atom and the rest electron mass (9.11×10^{-31} kg), respectively. E is the incident electron energy and c is the speed of light. E_{max} for Fe and Pd at 200 kV and 300 kV is summarized in Table R1.

Table R1. Maximum energy E_{max} transferred from the electron beam to Fe and Pd atoms.

	M	200 kV	300 kV
Fe	9.27×10^{-26} kg	9.40 eV	15.25 eV
Pd	1.77×10^{-25} kg	4.93 eV	8.00 eV

The threshold energies (E_d) of Pd and Pd metals have been measured to be 20 eV

and 34 eV, respectively (*Physica status solidi (a)*, 1979, 56(1): 157-168; *Journal of non-crystalline solids*, 2012, 358(3): 502-518). Experimental E_d for FePd has not been reported. However, it has been reported that E_d is roughly related to the melting temperature T_m by the equation $E_d = 0.0076 \times T_m + 9.8441$. Using FePd $T_m = 1577.15$ K in the phase diagram (*Journal of optoelectronics and advanced materials*, 2010, 12(9): 1869), E_d is estimated to be 22.8 eV. All these values are larger than the calculated E_{max} , confirming that the electron beam causes negligible beam damage. We have included this discussion in the revised manuscript. Detailed calculation is included in Supplementary Note 3. Please see page 6, lines 158-160.

J. Page 10, lines 257-259. "...indicating that the local bonding charge density is not as sensitive to the local chemical environment compared to atomic vibrations" is a very poor conclusion which does not make physical sense. One can only make such statements if one has subtracted the IAM structure factor from the measured structure factors, then the differences between the two sets of measurements will be put in the context of bonding electron (or difference) structure factors and not total electron density structure factors. You cannot make claims about bonding if you have not removed the IAM from your structure factors.

Reply: We would like to thank the reviewer for pointing this out. The comparison is indeed for the difference between $F_g^{EXP} - F_g^{IAM}$ but the axis labels and the legends are confusing. We have changed the x-axis labels to "(111)" and "(200)", and the y-axis label to " $F_g^{EXP} - F_g^{IAM}$ ".

K. Table 1, Page 10: What is the uncertainty associated with all the individual measurements? It is not sufficient to provide measurements from each CBED pattern without gauging the uncertainties of the individual measurements. There are many different ways to do this, but it is not clear that authors have determined local (individual) uncertainties.

Reply: The standard deviations of the refined parameters are evaluated using the method described in the paper by Tanaka and Tsuda (*Acta Crystallographica Section A: Foundations of Crystallography*, 1995, 51(1): 7-19). The error is propagated from the experimental CBED intensity to the refined parameters through the Jacobian, which describes the derivative of pixel intensity with respect to a fitting parameter. After the final iteration, the Jacobian is usually very small, and the resulting standard deviation of the refined parameters is negligible. For example, the standard error estimated using this method for F_{111} and F_{200} is 0.000424 and 0.000423, respectively, for the CBED pattern in Fig. 2d if F_{111} and F_{200} are relaxed during QCBED refinement. Therefore, according to the nonlinear fitting algorithm, the fitting results are very precise. The accuracy of the fitting is, however, difficult to determine because there is no known reference. Practically, we are more interested in accuracy than precision. A common approach we have been using is comparing fitting results from CBED patterns with different thicknesses as reported in the manuscript. We have added relevant discussion to the Methods section. Please see page 18, lines 451-452.

L. The authors discuss atomic displacements and local lattice distortions in a quasi-random $4 \times 4 \times 4$ supercell on Page 11, lines 276-288. It is not clear from this paragraph as to how the lattice parameters were relaxed or the atomic positions were relaxed. No mention has been made of DFT so far or in this paragraph. See point 24.

Reply: We have included more details on how the supercell was relaxed using DFT in the Methods. Please see pages 17-18, lines 427-434.

M. Page 11, line 291: You state that F_g starting from 220 onwards are not included in the refinement but this contradicts the statement that you did refine F_{220} on page 8, line 213. This relates to an earlier question as to why the refined values of F_{220} were not plotted in Fig. 3. There is a reference in the text on page 8 to two refinements being performed consecutively and the first is discussed, involving F_{220} but I cannot see any clear reference to a discussion of the second refinement. If there is a lot of material

discussed between the first and second refinements, then the authors should rewrite the text so that it is clear as to which one is being discussed and attention should be drawn back to the refinement overview (2 refinements) when commencing discussion of the second refinement if this is indeed what is being discussed on page 11. This needs to be made a lot clearer.

Reply: We agree with the reviewer that the refinement process needs to be clarified. As mentioned in the reply to the reviewer's comment 'H', for each CBED pattern, five refinements were performed consecutively by adding one more low order structure factor each time. As the following several comments are all about the role of F_{220} in the refinement process and the construction of bonding electron density, it is important to clarify why F_{220} is not included in the calculation of the deformation electron density. F_{220} has been included in the refinement, but after careful evaluation, we decided that it is better to exclude F_{220} for the construction of bonding electron density, for the following reasons.

First, the inclusion of F_{220} and later F_{113} does not noticeably reduce R_w , indicating that either the QCBED refinement is not sensitive to these two structure factors, or F_{220} and F_{113} are indeed close to IAM values. Using the refinement that relaxes four low-order structure factors up to F_{113} , the difference between F_g^{EXP} and F_g^{IAM} is summarized in Figure R22 for F_{111} , F_{200} , F_{220} , and F_{113} . F_{111} and F_{200} deviate from IAM values as all the measured data points are consistently on one side of the x-axis. For F_{220} and F_{113} , the data points are closer to the x-axis, and both of them are not different from IAM value considering the error bar. Therefore, it is possible that F_{220} and F_{113} are influenced by bonding, but capturing minute deviation from IAM values for structure factors after (200) is difficult.

Figure R22 Difference between experimentally measured structure factors and IAM structure factors.

Second, the scale factor $\frac{c\Omega s^2}{\gamma}$ used in Mott formula to convert electron diffraction structure factor U_g to X-ray diffraction structure factor F_g^X can magnify errors for large s . Therefore, any small error in the measurement of F_{220} can cause significant errors when constructing the bonding electron density map. The error bar for F_{111} , F_{200} , F_{220} , and F_{113} are 0.01, 0.02, 0.07, and 0.07, respectively, indicating that higher order structure factors have much larger measurement error. That is another reason why high order structure factors are excluded. A similar case can be found in the literature (*Science*, 2011, 331(6024): 1583-1586), where Nakashima et al. also use F_{111} and F_{200} to construct the deformation electron density map for FCC Al.

Third, the refinement results indicate that adding more structure factors in the refinement can deteriorate refinement results. **Figure R23** shows that when F_{113} is included, the error bars for B_{iso} , F_{111} , and F_{200} increase, but the averaged values do not change that much. Adding one more parameter to the refinement process encourages the other parameters to vary more freely, which makes it more difficult to find a global minimum. The tables and figures for fcc FePd in the main manuscript are therefore based on the refinement using, B_{iso} , F_{111} , and F_{200} . The values of F_{220} are from the refinement using B_{iso} , F_{111} , F_{200} and F_{220} . We have clarified this part in the main manuscript. Please see page 9, lines 226-241; page 10, lines 258-260.

Figure R23 Fitting results (a) B_{150} (b) F_{111} (c) F_{200} (d) F_{220} of 8 CBED patterns. The refinement was performed by incrementally including one more low order structure factor.

N. Further to the leaving out of F_{220} in the refinements discussed on p11, lines 291-293, to assume that it is reasonable to leave this structure factor out of refinements because it has assumed the IAM value in the cases of pure elemental FCC metals is naïve to say the least. The dissimilar electronegativities of Fe and Pd would make it unlikely that bonding is as delocalised as in pure elemental metals, and therefore, the assumption that F_{220} will assume the IAM value is highly questionable.

Where is the evidence that the authors can present for FePd to justify this approach?

Reply: We agree with the reviewer that more details need to be provided. As mentioned in the reply to comment ‘M’, the refined F_{200} values from 8 CBED patterns are not significantly different from IAM value within the standard deviation. As a result, we are not very confident in using F_{200} to construct the bonding electron density map. **Figure R24** compares the bonding electron density maps with and without F_{200} . In both cases, bonding electrons tend to accumulate at octahedral interstitial sites. We have explained that in the revised manuscript. Please see page12, lines 305-306.

Figure R24 Deformation electron density (a) with and (b) without F_{220} .

THE BEST TEST FOR THIS IS TO COMPUTE STRUCTURE FACTORS USING DFT FOR FePd AND COMPARE THEM TO THE IAM. THIS WILL GIVE SOME SORT OF AN ESTIMATE AS TO HOW MANY F_{hkl} ARE AFFECTED BY BONDING. RECENT STUDIES SHOW THAT IN INTERMETALLICS, 20-25 STRUCTURE FACTORS (SYMMETRY UNIQUE) ARE BONDING AFFECTED. TRUNCATION TO THE LOWEST 2 STRUCTURE FACTORS IN $\sin\theta/\lambda$ IN THIS WORK IS ALMOST CERTAINLY NOT GOING TO GIVE AN ACCURATE ASSESSMENT OF BONDING IN AN INTERMETALLIC. LOOKING AHEAD TO THE END OF THE PAPER, IT SEEMS THAT THE MAPS FROM DFT ALSO NEGLECT HIGHER ORDER STRUCTURE FACTORS BEYOND F_{200} . TO COMPUTE SUCH A HIGHLY TRUNCATED FOURIER SERIES WHEN SO MANY MORE TERMS ARE SURELY GOING TO BE NON-ZERO IS A MAJOR FLAW.

Reply: We agree with the reviewer that bonding could influence much more structure factors than just 2 low order structure factors. However, currently it is still very difficult to use any experimental methods including QCBED and X-ray diffraction to measure the minute deviation from IAM values for high order structure factors. Hopefully, the situation can be improved with the help of direct electron detectors, optimization of beam-sample orientation, and better refinement algorithm to choose F_g -sensitive disc

regions. This is the research direction of our group in the next couple of years.

O. Page 11, line 295: What do the authors mean by "flatten the electron density". This needs to explained carefully.

Reply: We agree with the reviewer that this sentence needs to be clarified. This is based on the observation that the experimental low order structures are lower than IAM values for pure elements such as Al (*Science*, 2011, 331(6024): 1583-1586), Cr, Fe, Co, Ni, Cu (*The Journal of Chemical Physics*, 2013, 138(8)). When two atoms are bonded, the bonding electron density becomes smoother or ‘flattened’, which can lower their Fourier transform, i.e., the structure factors. We have changed the sentence to “This is consistent with the typical trend that bond formation and charge transfer lead to a smoother electron density distribution and a reduced low-order F_g ” in the revised manuscript. Please see page 12, line 298.

P. Fig. 4, page 12: The deformation electron density plots are almost certainly incorrect because it is highly likely that bonding-affected structure factors have been ignored. Refer to point N above.

Reply: As mentioned in the reply to previous comments regarding F_{220} , it is indeed very difficult to measure those high order structure factors. To avoid confusion, in the revised manuscript we clearly defined that the deformation electron density map is calculated using F_{111} and F_{200} .

Q. Page 13: All discussions of bonding electron densities relating to Fig. 4 are invalid as the authors have probably ignored a large number of bonding-affected structure factors. See point N above.

Reply: We agree with the reviewer that it is probably inappropriate to call this bonding electron density. The term has been changed to deformation electron density in the revised manuscript.

R. Section 2.4: The whole section neglects any mention of how the bonding electron density was determined from DFT. This leaves one to assume that the same method as in the previous section was applied which was to only consider F_{111} and F_{200} and their symmetry equivalents. As per point N above, this approach is wrong.

Reply: We agree with the reviewer that more details need to be discussed on how deformation electron density was determined from DFT. The deformation electron density was calculated using low order structure factors up to $\frac{\sin\theta}{\lambda} = 0.263$, which is equivalent to F_{200} of fcc FePd. The detailed values of the structure factors for different supercells are now included in Supplementary Table 7-11. Generally, more low order structure factors are included due to larger unit cells. We have added more discussion in the revised manuscript. Please see page 15, lines 358-360.

Continuing with this point and point N, even table 2 suggests that only refining F_{111} and F_{200} is a flawed approach because whilst there is little difference between the QCBED and DFT results for F_{111} , there is actually a statistically significant difference between the QCBED and DFT results for F_{200} , implying that there is some compensation occurring due to the neglect of higher order structure factors deviating from the IAM.

Reply: This is a very interesting question. This is related to the reply to comment ‘N’, where we plot fitting results of 8 CBED patterns for refinements performed using the different number of low order structure factors. The reviewer is correct that there could be compensation between structure factors during the QCBED refinement (Figure R23). When F_{200} is included in the refinement, F_{111} decreases from 27.276 to 27.181 (see Supplementary Table 2). However, including F_{220} or F_{113} does not cause much change for the mean values of F_{111} or F_{200} . Instead, including F_{220} or F_{113} in the refinement increases the uncertainty of F_{111} or F_{200} (Figure R23). Therefore, the difference between experimental and theoretical F_{200} could be from other sources, such as *d*-electrons in Fe and Pd. Using the correct background correction and PSF,

F_{111} from thick regions (27.179) now matches much better with that from thin regions (27.183) (Supplementary Table 2), but both deviate from DFT value, suggesting that the excellent match between CBED and DFT in the previous version was a coincidence. We have added more discussion in the revised manuscript. Please see page 14, lines 354-357.

At the bottom of page 14, the use of the AIM method implies that the total charge density (all structure factors) is being used in the net charge analysis. This would contradict the treatment used so far which is to only consider F_{111} and F_{200} . Furthermore, the AIM net charge analysis is explained by the differences in electronegativities between Fe and Pd, which is precisely the point I made in point N above. So, the authors own conclusions contradict the exclusion of all structure factors apart from F_{111} and F_{200} and their symmetry equivalents. Is this not an obvious self-contradiction of this work?

Reply: We agree with the reviewer that the AIM analysis needs more explanation. WIEN2K uses full electrons for DFT calculation. Therefore, the full electron density map was calculated for all the supercells using WIEN2K. The AIM analysis was performed directly using the full electron density from WIEN2K. We have clarified this point in the revised manuscript. Please see page 15, lines 377-379.

S. Lines 376 and 377: The authors tie up their explanation with the argument of elemental FCC Fe. But iron in its elemental ground state is BCC. A DFT calculation will verify this. This is also a significant self-contradiction.

Reply: We agree with the reviewer that this needs some explanation. DFT calculation indicates that the formation energies of the bcc and fcc structures are -8.238 eV/atom and -8.078 eV/atom respectively, confirms that the bcc structure is more stable than fcc. The main point in the manuscript, is that if an Fe atom is surrounded by 12 Fe atoms as in fcc Fe, then the charge transfer is almost zero. We have changed the sentence to “Extrapolation of the linear relationship to the limiting case, where a Fe atom is

surrounded by 12 NN Fe atoms, predicts a vanishing net charge, in excellent agreement with the scenario of charge neutrality in pure metallic FCC Fe” in the revised manuscript. Please see page 16, lines 382-385.

In conclusion, I think this work has great potential but all of the fundamental flaws discussed above need to be addressed and fixed before I can recommend acceptance. This should not be too difficult for the authors to do – all they need to remember is to make their analysis physically sensible on all fronts and provide all the necessary evidence to support the approximations made. Some approximations that have been made are clearly self-contradictory and ignore the evidence right in front of the authors.

I am happy to referee this work again until it is acceptable for publication.

Reply: We would like to thank the reviewer for the insightful comments on improving the manuscript.

Reviewer #2 (Remarks to the Author):

This paper by Lin et al. addresses a very timely problem, that is the degree of disorder and impact on bonding in intermetallic alloys, which is very much relevant to recent interest in complex alloys, including high-entropy alloys. The approach that the authors use is quantitative convergent-beam electron diffraction (QCBED) and the system being studied here is FePd. While QCBED is a well-established diffraction technique, its applications for disordered crystals have been limited. Using QCBED, the authors have performed refinements of the disorder parameter, isotropic Debye-Waller factors (DWF) and the structure factors of (111) and (200) structure factors. The authors concluded that (1) disorder in FePd increases DWF, (2) has negligible influence on the bonding electron density, and (3) QCBED matches DFT and thus can be used to understanding bonding in chemically disordered systems.

While claim (1) is reasonable and expected, the claim (2) might be the result of limitations in the way the structure factors are probed, which needs further qualification. Claim 3 is encouraging and may further stimulate future studies on similar systems.

As written, the manuscript also contains several issues that need to be addressed.

Reply: We appreciate the reviewer's very positive comments on our work. The comments by the reviewers are addressed point-by-point as follows.

First, the structure factors in equations 2 and 3 ignores the imaginary part from the electron absorptive potential. This part is sensitive to thermal vibrations and thus the temperature dependent part of DWF, while the static part from disorder should have very little effect. Thus, how absorption is dealt with is critical for the refinement.

Reply: The reviewer is correct in speculating that absorption plays an important role in QCBED refinement. For all the refinement performed in the manuscript, if a structure factor is relaxed, the real and imaginary parts are independently relaxed. The initial values of the imaginary part are based on this reference (*Acta Crystallographica Section*

A: *Foundations of Crystallography*, 1990, 46(3): 202-208), while the refined results are listed in Table R2 below. We have clarified the treatment of absorption factors in the revised manuscript and included fitting results also in Supplementary Table 5. Please see page 6, line 166; page 11, line 260; page 18, line 454.

Table R2. The imaginary part of the electron diffraction structure factors from the literature and QCBED refinement

	U_{111}	U_{200}	U_{220}
Bird & King	0.00556	0.00543	0.00500
EXP	0.00652	0.00680	0.00515

Second, line 48, “disorder has proven very difficult as their interaction with the incident electron beam becomes increasingly weaker.” This needs a clarification; it seems what the authors mean here is the decreasing intensity of order sensitive reflections.

Reply: The reviewer is correct that this sentence means that the intensity of order-sensitive reflections decreases for charge, orbital, and spin ordering, which makes them difficult to capture using conventional imaging or diffraction techniques. We have changed this sentence to “probing local disorders of charge, orbital, or spin remains challenging as their diminishing effects on measurable parameters such as electron density, electrostatic potential, and ordered-sensitive diffraction intensities” in the revised manuscript. Please see page 2, lines 48-50.

Third, the authors have used relatively thick samples with thickness greater than 200 nm and reach ~400 nm. Even though the probe used is small (0.5 nm), the interaction volume is large and thus the measured structure factors is averaged over many unit cells, which is probably while the reported study is not sensitive to fluctuations in bonding charge density that may arise from disorder.

Reply: This is a very interesting question that we did not clarify in the manuscript. The CBED pattern acquired from thin regions of the disordered FePd sample tends to show

symmetry breaking in the CBED discs, which significantly deteriorates refinement results. Symmetry breaking is caused by a local strain field resulting from bending or defects. The FePd sample tends to severely bend at thin regions, likely due to jet polishing. Moreover, there are lots of defects, such as dislocations and planar defects. Therefore, for practical reasons, most CBED patterns refined in this manuscript are from relatively thick regions. However, the results suggest that thickness does not influence the refinement results such as the disorder parameter (Fig. 2e), DWF (Fig. 3e), or low order structure factors (Fig. 3f-h). The fitting parameters as a function of thickness are also summarized in Figure R25.

Figure R25 The influence of thickness on different fitting parameters

Fourth, the use of bonding electron density in the title implies a measurement of charge density, which is not the case here. The structure factors measured here are sensitive to bonding, but alone are not sufficient to describe the full 3D charge density. A modification is recommended here. Second, the disorder parameter in the title has a very specific meaning, e.g. in an intermetallic alloy, which should be made clear.

Reply: The reviewer is correct that it is a little bit misleading to call this bonding electron density because it is difficult to measure all the bonding-influenced structure factors. Therefore, we have to assume that high order structure factors are identical to IAM values. This has been thoroughly discussed in the reply to the first reviewer's comments. To avoid confusion, in the revised manuscript, "bonding electron density" has been replaced by "deformation electron density".

The disorder parameter defined here is based on a conventional definition of long-range order parameters S (*physica status solidi (a)*, 1988, 110(1): 77-82). If we use tp2 cell as an example, the long-range order parameter of Fe at (0 0 0) is defined as $S = \frac{Occ.(Fe_{Fe}) - F_{Fe}}{1 - F_{Fe}}$, where $F_{Fe} = 0.5$ is the atomic fraction of Fe in FePd, and $Occ.(Fe_{Fe})$ is the occupancy of Fe at (0 0 0), which is related to $Occ.(Fe_{Pd})$ as follows,

$$Occ.(Fe_{Fe}) + Occ.(Fe_{Pd}) = 1$$

Therefore, $S = 1 - 2 * Occ.(Fe_{Pd}) = 1 - \eta$. We have explained the definition of η in the revised manuscript (please see page 4, line 100) and included the discussion in the Supplementary information as Supplementary Note 1.

Fifth, it is not clear how to make sense of the comparison to DFT calculations. The supercells used for calculation no longer have the fcc lattice symmetry, is the fcc structure factor sufficient to describe the charges in a supercell? Does reflections along different directions differ? The results here also seem to not confirm any particular bonding configuration, which is in direct contradiction to the abstract claim that they revealed and confirmed the local bonding environment.

Reply: We agree with the reviewer that this needs to be clarified. The supercells generally have lower symmetry, and therefore all low order structure factors with $\frac{\sin\theta}{\lambda}$ less or equal to the (200) reflection ($\frac{\sin\theta}{\lambda} = 0.263 \text{ \AA}^{-1}$) in fcc FePd are included to

construct the deformation charge density. The low order structure factors of each cell used in the manuscript are now included in Supplementary Table 7-11. The reviewer is correct that F_g along different directions could differ, but are very close to each other. This suggests that although Fe and Pd do have some charge transfer as indicated by AIM, the behaviour is still quite isotropic.

We agree with the reviewer that the results suggest that the structure factors are not sensitive to the local bonding environment. This might be special for metallic bonding in metals and intermetallics. In the revised manuscript we added some discussion on this topic “This unexpected agreement suggests that the dominant charge redistribution in FePd arises from global symmetry and coordination effects rather than site-specific chemical disorder”. Please see page 14, lines 334-336. Moreover, the deformation electron density map, however, combined with DFT, can to some extent reveal where the bonding electrons are located. We have downplayed this sentence in the revised manuscript to “This study establishes QCBED as a robust method for quantifying local disorder parameters in chemically disordered systems, bridging a critical gap in the characterization of disordered materials”. Please see page 2, lines 33-35.

Other minor comments:

Line 48-51 This line is potentially misleading, since disorder also induces atomic displacements, which also impact structure factors and could dominate over the electronic structure part.

Reply: We thank the reviewer for pointing this out. It has been shown that the coupling between lattice and charge, orbital, or spin could lead to lattice displacements that can be captured by STEM. According to this equation $F_g^X = \sum_i f_i(s) e^{-B_i s^2} e^{-2\pi i g \cdot r_i}$, lattice displacements can cause changes in structure factors, which is beneficial for CBED measurement. We have changed this sentence to “all forms of disorder perturb electronic structures or even lattice displacements, which collectively modify bonding

and electron density distribution” in the revised manuscript. Please see page 2, lines 50-52.

Line 79 - The way this is written makes it sound like it is the first time the DWF and structure factor have been refined. This is certainly not true, it might be the first time the disorder parameter has been refined.

Reply: The reviewer is correct that this sentence needs to be revised. We have removed “for the first time” to avoid confusion. Please see page 3, lines 82-84.

Lines 164-182 These R_w values seem high and should be compared with other metrics such as chi-square and the standard R factor. Additionally, the R_w is calculated with the knowledge of intensity uncertainty (sigma). How this was obtained should be specified in the methods. Related to this, how the intensity recorded after a GIF on CCD is processed should also be described, for example, the method used to correct the MTF or PSF of CCD.

Reply: We agree with the reviewer that more details need to be provided. In the revised manuscript, we have significantly reduced the R_w values for CBED patterns acquired from thick regions of disordered FePd, by optimizing the PSF (see the reply to the first reviewer’s comment ‘C’), removing the background, and setting the correct values for the noise or intensity uncertainty. At the same time, the refinement results using this optimized refinement procedure are less sensitive to thickness change, which certainly makes more sense. We would like to thank both reviewers for their comments on improving the refinement results. The intensity uncertainty σ_i is defined as $\sigma_i^2 = \sigma_{ip}^2 + \sigma_{bkgd}^2$, where $\sigma_{ip} = \sqrt{I_i^{obs}}$ and σ_{bkgd} is a constant background value extracted from the region between discs. Besides the weighted R_w value defined as,

$$R_w = \left(\frac{\sum_i \frac{(I_i^{obs} - I_i^{cal})^2}{\sigma_i^2}}{\sum_i \frac{(I_i^{obs})^2}{\sigma_i^2}} \right)^{\frac{1}{2}},$$

the standard R factor defined as $R = \frac{\sum_i |I_i^{obs} - cI_i^{cal}|}{\sum_i I_i^{obs}}$, has also been calculated for each refinement. We have added the R value for the eight refinements in Table 1.

The refinement minimizes S which is defined as $S = \sum_i \frac{(I_i^{obs} - cI_i^{cal})^2}{\sigma_i^2}$. The details are now included in the revised manuscript.

Line 250 - There are several mentions of the probe size, but equally important is the convergence angle. This should be added to Table 1 to know how much of sample volume is covered.

Reply: The convergence angle has been measured using the disc size and the distance between discs. For spot size 2.4 nm, using the datapoints in Figure R16-17, the distance between (000) disc and (001) disc is $R_1 = 263.2$, and the disc radius is 222 pixels. $\frac{\sin\theta}{\lambda}$ of the (001) disc is 0.1313 \AA^{-1} , corresponding to 3.3 mrad for $\lambda = 0.0251 \text{ \AA}$. The convergent angle is, therefore, $2 \times 3.3 \times \frac{222}{263.2} = 5.6 \text{ mrad}$. The convergent angle for spot size 0.5 nm is the same. We have added the values of convergence angle in the revised manuscript as suggested by the reviewer. Please see page 10, line 256.

Reviewer #3 (Remarks to the Author):

Reply: We would like to thank the reviewer for reviewing this manuscript.

Responses to Comments of NCOMMS-24-59352A

We would like to thank the reviewers for the professional and valuable comments on our manuscript. The manuscript has been revised according to the comments of the reviewers.

REVIEWER COMMENTS

Reviewer #1 (Remarks to the Author):

I am impressed with the rigour of the corrections that have been made and the explanations that have been directed to each of my points made in round 1 of refereeing. I thank the authors for their courtesy and care in responding.

The main issue that I had which centred on the assumption that F_{111} and F_{200} were enough to characterise the bonding electron density in FePd has largely been tackled and I am also happy that the authors have changed "bonding electron density" to "deformation electron density". This is more acceptable.

Reply: We thank the reviewer for the encouraging feedback and recognition of the effort put into the revision. We are particularly glad that the detailed analysis regarding the use of F_{111} and F_{200} , along with the adoption of the term “deformation electron density”, has resolved the primary concerns from the initial review. We are grateful for the reviewer’s careful consideration and valuable input, which has undoubtedly strengthened the paper.

Some comments to follow up on your responses:

- I thought that superlattice reflections due to weak ordering might be present and you showed this nicely with the SAD pattern in Fig. R2. They are very weak indeed and so I can accept that they will make a minimal contribution. This has been quite nicely demonstrated.*

- *The approach of dealing with the PSF by convoluting the calculated patterns instead of deconvoluting the experimental patterns is very inventive and I like this approach. However, YOU USED A CCD ON A TRIDIEM GIF. This presents a significant oversight, i.e. that the PSF associated with CCDs has a very large and significant tail and that this is best modelled by a Lorentzian. In fact a standard multicomponent decomposition reveals that the Lorentzian is very significant and therefore the only sensible way of modelling a CCD PSF is the weighted combination of Gaussians and Lorentzians.*

If I may be so bold as to suggest a simple solution for this issue: clearly, it is difficult to have a two-parameter optimisation of the Gaussian-Lorentzian function because of the nature of your piece-wise approach to the refinements. An easy way out of this is to measure your detector PSF accurately by the large range of simple methods available out there and then convolute the smoothed experimentally determined PSF with your calculations. This will negate you having to adjust two parameters in the PSF convolution step.

- *To claim that distortions in the ZOLZ are linear is a first order approximation and it is good if you state this in the adjusted manuscript.*

- *Noise in CCD data is not just constant + Poisson. There is a huge addition due to gain noise since during each exposure, the detector itself changes state from the state it was in when it was gain referenced. Gain correction noise is linear and adds a significant fraction to the overall noise characteristics, especially when frame averaging (eg. Zuo, Ultramicroscopy 1996). You really need to say something about this or better still, do something about this.*

- *My main concern has been very well taken care of by Figs R22, R23 and R24. I will put this issue out of my mind as long as you include those figures and an associated explanation in the Supplementary Information.*

Reply: We thank the reviewer again for the careful reading and insightful follow-up

comments. We are very pleased that our previous response and revision successfully addressed the reviewer's main concerns.

IN SUMMARY:

I am happy to recommend acceptance, ON THE CONDITIONS:

- *Please use a mixed Gaussian/Lorentzian OR experimentally measured PSF in all your pattern-matching analyses. The latter is better as it can account for detector anisotropy by not assuming that the PSF is radially symmetric.*

Reply: We thank the reviewer for this insightful suggestion regarding the PSF. We agree that accurately modelling the PSF is significant, particularly for future high-precision measurements. To address this comment thoroughly, we have systematically investigated the influence of the PSF model on our refinement results and characterized the experimental PSF.

We compared refinement results using three analytical PSF models: a Gaussian distribution $G(x; \sigma)$, a Lorentzian distribution $L(x; \gamma)$, and their convolution, the Voigt distribution $V(x; \sigma; \gamma)$. These are defined as:

$$G(x; \sigma) \equiv \frac{e^{-\frac{x^2}{2\sigma^2}}}{\sqrt{2\pi}\sigma}$$

$$L(x; \gamma) \equiv \frac{\gamma}{\pi(\gamma^2 + x^2)}$$

$$V(x; \sigma; \gamma) \equiv \int_{-\infty}^{\infty} G(x'; \sigma)L(x - x'; \gamma)dx'$$

The full width at half maximum (FWHM) for the three distributions are:

Gaussian distribution: $f_G = 2\sqrt{2\ln(2)}\sigma$;

Lorentzian distribution: $f_L = 2\gamma$;

Voigt distribution: $f_V = 0.5343 \times f_L + \sqrt{0.2169f_L^2 + f_G^2}$ (*Journal of Quantitative Spectroscopy and Radiative Transfer*, 1977, 17(2): 233-236).

We first optimized the fits for the representative CBED pattern in Fig. 3 using pure Gaussian and pure Lorentzian PSFs (grid size 101). Using a Gaussian PSF, the minimum R_w of 0.2042 was achieved at $f_G = 1.67$ pixels (Figure R1a). Using a Lorentzian PSF, the minimum R_w was slightly lower at 0.2004, achieved at $f_L = 0.38$ pixels (Figure R1b).

Figure R1 R_w value as a function of (a) Gaussian FWHM (f_G) and (b) Lorentzian FWHM (f_L) used in the PSF convolution (grid size 101).

Next, we performed refinements using the Voigt distribution, varying both σ and γ (equivalently, f_G and f_L). We mapped the resulting R_w values (Figure R2). The global minimum $R_w = 0.2002$ was found using a Voigt profile with $f_G = 1.06$ pixels

and $f_L = 0.3$ pixels. This represents only a marginal improvement over the pure Lorentzian fit. The contour map shows a relatively flat minimum, indicating that similar R_w values are obtained for a range of f_G and f_L combinations.

Figure R2 Optimization of Voigt PSF parameters. Contour map shows R_w value as a function of the Gaussian FWHM f_G and Lorentzian FWHM f_L components of the Voigt profile used for convolution.

We then evaluated the impact of these different optimal PSFs (Gaussian, Lorentzian, Voigt) on the key refined parameters (Table R1). Despite the small differences in R_w , the refined structure factors (F_{111} and F_{200}) vary by only ~ 0.003 e⁻/atom across the three PSF models, corresponding to a negligible difference of $\sim 0.01\%$. Other parameters like thickness and the DWF also show minimal variation.

Table R1 The refinement results corresponding to the minimal R_w value when using PSFs of Gaussian, Lorentzian, and Voigt distributions.

Parameters	Gaussian	Lorentzian	Voigt
f_G	1.67		1.06
f_L		0.38	0.3
f_V			1.23
R_w	0.2042	0.2004	0.2002
R	0.1934	0.1896	0.1893
Thickness (Å)	4386.2	4384.7	4385.0
B_{iso} (Å ²)	0.472	0.484	0.481674
F_{111} (e ⁻ /atom)	27.166	27.169	27.168
F_{200} (e ⁻ /atom)	25.545	25.548	25.548

To further assess robustness, we analysed the refined parameters within the region where $R_w < 0.205$ using the Voigt PSF (Figure R3). Within this region of good fits, the refined parameters remained remarkably stable: DWF: $0.4832 \pm 0.006 \text{ \AA}^2$ (mean \pm std. dev.); F_{111} : $27.169 \pm 0.002 \text{ e}^-/\text{atom}$; F_{200} : $25.548 \pm 0.001 \text{ e}^-/\text{atom}$. This confirms that the refinement results are insensitive to the precise shape of the analytical PSF used, as long as its width is reasonably optimized.

Figure R3 Surface plots (top row) and contour maps (bottom row) show the refined (a, b) B_{iso} , (c, d) F_{111} (e^-/atom), and (e, f) F_{200} (e^-/atom) as a function of the Voigt components f_G and f_L . The white dashed lines delineate the region where $R_w < 0.205$.

The preceding analysis demonstrates that the choice of analytical PSF (Gaussian, Lorentzian, or Voigt), once optimized, does not significantly influence the key refined parameters like structure factors within the precision required for this study. We subsequently re-refined all 8 experimental CBED patterns using the optimized Voigt distribution. These results are summarized in Table R2. The resulting structure factors (red font) are nearly identical to those previously obtained (black font) using the Gaussian distribution. While the refined DWF showed minor adjustments, these differences were confirmed to have no significant impact on the interpretation of the derived deformation electron density maps.

Table R2 The refinement results for the 8 CBED patterns in the manuscript.

Image index	Spot size (nm)	Thickness (Å)	R_w	R	B_{iso} (Å ²)	F_{111} (e ⁻ /atom)	F_{200} (e ⁻ /atom)	F_{220} (e ⁻ /atom)
1	0.5	2642.0	0.1762	0.1705	0.514	27.182	25.502	21.457
		2641.1	0.1770	0.1710	0.524	27.181	25.499	21.448
2	0.5	2640.4	0.2101	0.2013	0.430	27.168	25.520	21.356
		2639.9	0.2055	0.1967	0.436	27.167	25.517	21.355
3	0.5	2670.2	0.1760	0.1654	0.485	27.182	25.535	21.410
		2669.4	0.1751	0.1641	0.491	27.182	25.531	21.403
4	0.5	2652.4	0.2366	0.2164	0.505	27.200	25.499	21.555
		2651.2	0.2368	0.2160	0.517	27.198	25.492	21.548
Mean					0.48(4)	27.18(1)	25.51(1)	21.44(8)
					0.49(4)	27.18(1)	25.51(2)	21.44(8)
5	2.4	4386.2	0.2042	0.1934	0.472	27.166	25.545	21.415
		4384.9	0.2002	0.1893	0.482	27.168	25.548	21.400
6	2.4	4177.8	0.1893	0.1809	0.464	27.178	25.516	21.423
		4177.8	0.1863	0.1785	0.467	27.179	25.514	21.411
7	2.4	4026.9	0.2132	0.2041	0.481	27.180	25.543	21.533
		4025.8	0.2145	0.2044	0.489	27.182	25.544	21.516
8	2.4	4041.0	0.1824	0.1826	0.484	27.191	25.522	21.400
		4039.4	0.1812	0.1809	0.493	27.193	25.522	21.374
Mean					0.475(9)	27.18(1)	25.53(2)	21.44(6)
					0.48(1)	27.18(1)	25.53(2)	21.43(6)

We also experimentally characterized the PSF of our detector system using two methods. The first method measures the modulation transfer function (MTF) from a uniformly illuminated image and analysed its Fourier transform (Figure R4). The angular distribution of the FFT magnitude at various spatial frequencies was constant, confirming that our experimental PSF is highly isotropic, addressing a specific point raised by the reviewer.

Figure R4 Uniformly illuminated CCD image (Top Left) and the corresponding FFT (Top Right). Radially averaged profile of the FFT magnitude, representing the experimental MTF (Bottom Left). Angular distribution profiles (Bottom Right) of the FFT magnitude averaged over annuli at different radius, showing near-constant values indicative of a highly isotropic PSF/detector response.

Comparing the radially averaged profile (MTF) to the MTFs of the optimized analytical PSFs (Figure R5), we found the experimental MTF decays more slowly at higher frequencies. Here the analytical PSFs are scaled considering the sampling rate. This suggests that other experimental factors contribute significantly to the overall pattern blurring, as also suggested by previous comment on local lattice variation from the reviewer.

Figure R5 The experimentally measured MTF is compared to the MTFs calculated via Fourier transform of the optimized analytical PSFs determined from pattern fitting: Gaussian, Lorentzian, and Voigt.

The second method measured the line spread function (LSF) from an edge. We inserted a beam stop to create a sharp edge on the detector (**Figure R6**). Differentiating the line profile across the edge yields the LSF (**Figure R7**). Comparing this experimental LSF to the optimized analytical profiles showed that the LSF was broader than the optimal Lorentzian but narrower than the optimal Gaussian and Voigt profiles derived from pattern fitting. This difference further supports the idea that the convolution process in the refinement effectively models a combination of the detector PSF and other sample-related broadening effects.

Figure R6 (a) CCD image showing a sharp edge created by inserting a beam stop. (b, c) Intensity line profile averaged parallel to the edge, showing the intensity transition used to derive the LSF.

Figure R7 Comparison of experimental LSF and analytical PSFs.

In summary, we systematically tested Gaussian, Lorentzian, and Voigt PSFs. While the Voigt profile yields a marginally lower R_w (0.2002 vs 0.2042 for Gaussian), this improvement is minimal. The choice among these optimized analytical PSFs has a negligible impact on the refined parameters including the structure factors and DWF. Experimental characterization confirms the PSF is isotropic but indicates that the overall blurring observed in experimental CBED patterns arises from a combination of the detector PSF and other blurring effects. Therefore, while using a Voigt or experimentally measured PSF is ideal in principle, our analysis shows that for the current study, the specific choice of analytical PSF (within the optimized range) does not significantly affect the conclusions. The convolution effectively accounts for the dominant blurring effects.

We have updated the manuscript to reflect these findings. We now explicitly mention the use of Voigt distributions as an option in the main manuscript (page 6, lines 157-159), now reads: “For I_i^{cal} , overall blurring effects, including the point spread function (PSF) and potentially local lattice variations, are modelled by convolution with a two-dimensional (2D) Gaussian distribution or Voigt distribution”.

We have re-run the refinements to obtain structure factors and DWFs from the eight CBED patterns using the optimized Voigt distribution for consistency, and updated the relevant values in Table 1, Fig. 3c-h, and Fig. 4d. The updated figures are visually identical to the previous version.

Figure R2 (the Voigt R_w map) has been added to Supplementary Fig. 9 to provide supporting evidence for the robustness of the fit. We also added Supplementary Note 3 to describe the analytic PSFs used in the refinement. We believe that these investigations and revisions thoroughly address the reviewer's valid point.

• *Please account for linear gain noise in your analyses.*

Reply: We thank the reviewer for pressing the important point regarding linear gain noise in CCD detectors, and for the reference to the detailed work by Zuo et al. (*Ultramicroscopy*, 1996, 66(1-2): 21-33; *Ultramicroscopy*, 1996, 66(1-2): 35-47). We agree that a complete noise model should account for this effect.

Following Zuo's formulation, the noise variance in a pixel i can be described more comprehensively as: $\sigma_i^2 = \sigma_{ip}^2 + \sigma_{bkgd}^2 + \Delta(I_i^{obs})^2$, where Δ represents the contribution from the linear gain noise. Zuo et al. reported $\sqrt{\Delta} \sim 0.5\%$ for their characterized CCD. To evaluate the potential impact of this term on our specific results, we performed a sensitivity analysis. We re-ran the refinement for the CBED pattern in Figure 3, systematically varying the assumed value of $\sqrt{\Delta}$ from 0 (our original model) up to 0.01 which doubles the value reported by Zuo. The results are shown in Figure R8. As $\sqrt{\Delta}$ increases from 0 to 0.01, The standard R-factor remains remarkably stable, while the weighted R_w factor shows only a slight, gradual increase. The refined DWF and structure factors exhibit minimal changes. DWF varies from 0.4814 \AA^2 to 0.4816 \AA^2 (a difference of 0.0002 \AA^2). F_{111} varies from 27.1680 e⁻/atom to 27.1681 e⁻/atom (a difference of 0.0001 e⁻/atom). F_{200} varies from 25.5460 e⁻/atom to 25.5476 e⁻/atom (a difference of 0.0016 e⁻/atom).

Figure R8 The refinement results as a function of linear gain. (a) The weighted R_w factor. (b) The standard R factor. (c) Thickness. (d) DWF. (e, f) F_{111} and F_{200} .

This analysis demonstrates that even incorporating a conservatively high estimate for linear gain noise induces only negligible changes in the refined structure factors and DWF. These variations are smaller than those arising from other modelling choices (like the PSF details discussed previously) and do not affect the derived deformation density maps or the scientific conclusions drawn in this manuscript.

Accurately measuring the specific Δ for the detector at the time the historical data was acquired is unfortunately not feasible. Given that our sensitivity analysis shows a negligible impact of this term on our conclusions for this dataset, and the difficulty in precisely determining Δ retrospectively, we have retained the noise model with $\Delta = 0$ for the final refinements presented. We acknowledge that for future work, especially

for 4D-CBED that demands higher precision and longer exposure period, explicitly measuring and including Δ would be better.

We have updated the manuscript to clarify the noise model and justify our approach. The Methods section (page 18, lines 469-474) now includes the grain term in the definition of noise: “Here, σ_i represents the intensity uncertainty in the i^{th} pixel and is defined as $\sigma_i^2 = \sigma_{ip}^2 + \sigma_{bkgd}^2 + \Delta(I_i^{obs})^2$, where $\sigma_{ip} = \sqrt{I_i^{obs}}$, σ_{bkgd} is the constant background intensity, and Δ represents linear gain noise (*Ultramicroscopy*, 1996, 66(1-2): 21-33; *Ultramicroscopy*, 1996, 66(1-2): 35-47). Following a sensitivity analysis (Supplementary Fig. 16), the contribution from linear gain noise was found to have a negligible impact on the refined parameters and conclusions for this study and was therefore not explicitly included in refinement.” Figure R7 has been included as Supplementary Fig. 16.

• *Please include Figs R22, R23 and R24 and a covering explanation somewhere either in the text or Supp Info, to assure the reader that the deformation electron density you are presenting is a close representation of the true bonding ED.*

Reply: We appreciate the reviewer’s comment emphasizing the need to explicitly demonstrate that our presented deformation electron density is a close representation of the true bonding electron density. The relevant Figures R22, R23, R24 are indeed included in the Supplementary Information as Supplementary Figs. 13, 14, and 15. Recognizing that a covering explanation is beneficial, we have now revised the manuscript on page 12, lines 316-325: “ F_g^X s starting from F_{220} are excluded from the $\Delta\rho^{EXP}(r)$ construction because their experimental error bars are significantly larger than those of F_{111}^{EXP} and F_{200}^{EXP} (Supplementary Figs. 13-14). Consequently, these high-order reflections are indistinguishable from IAM values within the QCBED measurement error (Supplementary Table 4). This behaviour is consistent with previously observations for other FCC materials such as Al, Ni, and Cu. This suggests that $\Delta\rho^{EXP}(r)$ calculated using only F_{111}^{EXP} and F_{200}^{EXP} provides a close representation

of the true bonding electron density. Furthermore, a direct comparison of $\Delta\rho^{EXP}(r)$ generated with and without F_{220} confirms that F_{220} has a negligible influence on the result (Supplementary Fig. 15).”

I am happy to look at these points again once the manuscript has been suitably revised.

Bravo - I look forward to seeing this article in print - these changes should be fairly easy to implement.

Reply: We greatly appreciate your positive comments and encouragement! We have revised the manuscript accordingly.

Reviewer #2 (Remarks to the Author):

This revised version addressed the issues that were raised in my previous report. The authors' response is acceptable, except following which needs to be addressed:

Reply: We are grateful to the reviewer for the positive assessment of our revised manuscript. We appreciate the continued insightful feedback.

Page 2: Lines 48-52. For the sentence of “remains challenging as their diminishing effects on measurable parameters such as electron density, electrostatic potential, and ordered-sensitive diffraction intensities”, this is misleading, the disorder will increase diffuse scattering, but this is not measured here. The problem is associated with using Bragg diffraction alone to determine disorder effect, thus the problem is the sampling issue.

Reply: We thank the reviewer for highlighting this point. We agree that our original wording was misleading. The primary limitation is indeed the sampling issue associated with using Bragg diffraction to characterize disorder. To address this, we have revised the sentence on page 2, lines 46-49: “While atomic-resolution imaging and spectroscopy techniques offer insights into local chemical disorder, probing subtle local disorders related to charge, orbital, or spin remains challenging when relying solely on sampling ordered-sensitive Bragg diffraction intensities.”

In “Crucially, all forms of disorder perturb electronic structures or even lattice displacements”, “or even lattice” should be replaced by “and atomic”, since it is atoms displace around defects, that cannot be avoided.

Reply: We appreciate the reviewer’s suggestion for improved clarity. We have implemented this recommended change. The sentence on page 2, lines 49-51, now reads: “Crucially, all forms of disorder perturb electronic structures and atomic displacements, collectively modifying bonding and electron density distribution.”

Page 6: Lines 181-182/192-201. I'm not sure if I'm reading the contour plots in Fig.2j and Fig.2k incorrectly, but it looks like the minimized eta value is around 0.45 in the contour plots. These results either need an explanation or the previous values need to be reassessed considering relaxation of the DWF.

I think this is a major problem that needs to be addressed before publication. It is pretty clear the DWF is impacting the refinement. I think the plot needs to clearly state what the DWF used during refinement was. The contour plots clearly show local minima in the 0-0.5 range shown. Additionally, the R_w in these plots shows values at or below the determined minima for the refinement. It is possible that the graph is simply mis-labeled from 0-0.5 instead of 0-1.0, but this should be carefully checked.

Reply: We sincerely thank the reviewer for identifying this critical error in the labelling of the x -axis in Fig. 2i, 2j and 2k. The reviewer is correct that the x -axis labels were mislabelled in the original manuscript. The disorder parameter η ranges from 0 to 1. We apologize for this oversight, as we had inadvertently plotted the Fe occupancy on the Pd site (which ranges from 0 to 0.5) directly as the disorder parameter (η), while η is defined as twice the occupancy. We have replaced the figures with the correctly labelled versions in Fig. 2i-k of the revised manuscript and Supplementary Fig. 11 in the supplementary information.

Regarding the refinement process and the contour plots, the main refinements, which yielded the R_w values presented in Fig. 2e-h, involved optimizing DWF and other refinement parameters at fixed η ranging from 0 to 1 with a step size 0.02. The contour maps shown in Fig. 2i-k were calculated to visualize the R_w landscape around the refined minima and illustrate the sensitivity of the fit to both η and B_{iso} . These maps were generated using a coarser grid of fixed η values (from 0 to 1 with a step size of 0.1) and fixed B_{iso} values (from 0.1 to 1.1 \AA^2 with a step size of 0.1 \AA^2). The R_w value was calculated at each grid point. As the reviewer noted, the minima found on these discrete contour map grids may not correspond exactly to the minima found during the continuous refinement process, and the R_w values at the grid minima

should generally be equal to or slightly higher than the refined minima.

With the x -axis correctly labelled (0 to 1), for Fig. 2i, the minimum R_w value on this grid occurs at $\eta = 0$, $B_{iso} = 0.2 \text{ \AA}^2$, with $R_w = 0.17783$. This is slightly above the refined value of $R_w = 0.17778$ obtained at $\eta = 0$ (Fig. 2f), which is expected due to the discrete grid sampling versus continuous refinement. Similarly, for Fig. 2j, the minimum R_w value on this grid occurs at $\eta = 0.9$, $B_{iso} = 0.3 \text{ \AA}^2$, with $R_w = 0.1407$. This grid minimum R_w is slightly higher than the refined value of $R_w = 0.1382$ obtained at $\eta = 0.855$ (Fig. 2g). For Fig. 2k, the minimum R_w value on this grid occurs at $\eta = 0.8$, $B_{iso} = 0.4 \text{ \AA}^2$, with $R_w = 0.1814$. This grid minimum R_w is slightly higher than the refined value of $R_w = 0.1808$ obtained at $\eta = 0.856$ (Fig. 2h).

We have clarified this in the revised manuscript and reported the R_w values with an additional significant figure to ensure the precision accurately reflects these findings (Fig. 2f-h, Table 1, Supplementary Figs. 3-5). We have added the specific locations (η , B_{iso}) and R_w values of the minima found in the contour map grid in the revised manuscript to avoid confusion. Page 9 lines 194-205 now read: “The coupling between η and B_{iso} was then evaluated by generating the R_w isosurfaces on a grid of fixed η and B_{iso} values (η step: 0.1, B_{iso} step: 0.1), while all other parameters were relaxed during QCBED refinement (Fig. 2i-k). A clearly defined global minimum on the R_w isosurface for each of the three CBED patterns verifies simultaneous sensitivity to both η and B_{iso} . For the ordered sample (Fig. 2i), the global minimum on the grid ($R_w = 0.17783$) is found at $\eta = 0$ and $B_{iso} = 0.2 \text{ \AA}^2$, consistent with the refinement result ($\eta = 0$, $R_w = 0.17778$). For the disordered samples (Fig. 2j, k), the grid minima are found at high disorder parameters: $R_w = 0.1407$ at $\eta = 0.9$, $B_{iso} = 0.3 \text{ \AA}^2$ for 772.2 \AA thickness, and $R_w = 0.1814$ at $\eta = 0.8$, $B_{iso} = 0.4 \text{ \AA}^2$ for 4403.4 \AA thickness. These grid minima R_w values are slightly higher than the respective refined minima ($R_w = 0.1382$ at $\eta = 0.855$; $R_w = 0.1808$ at $\eta = 0.856$), as expected due to the discrete sampling.”

Reviewer #3 (Remarks to the Author):

Reply: We would like to thank the reviewer for the insightful comments.

Responses to Comments of NCOMMS-24-59352B

We would like to thank the reviewers for the professional and valuable comments on our manuscript.

REVIEWER COMMENTS

Reviewer #1 (Remarks to the Author):

Dear Authors,

Thank you for treating all of my comments with great rigour and depth. The sensitivity tests that you have done with respect to different PSFs and incorporation of linear noise are very convincing and I believe that you have solid conclusions that are worth being published.

Bravo - I like this paper and look forward to seeing it in print.

The amendments more than satisfy me and I am actually quite surprised with the latest depth of treatment.

I recommend acceptance in this latest form without any further changes.

Yours sincerely, referee 1.

Reply: We appreciate your time and effort in reviewing our manuscript throughout this process.

Reviewer #2 (Remarks to the Author):

I am satisfied with the revisions that authors have made in addressing the remaining issues pointed out in my last review.

Reply: We appreciate your time and effort in reviewing our manuscript throughout this process.

Reviewer #3 (Remarks to the Author):

Reply: We appreciate your time and effort in reviewing our manuscript throughout this process.